# Extended Agriculture-Vision: An Extension of a Large Aerial Image Dataset for Agricultural Pattern Analysis

**Jing Wu**                                                                *jingwu6@illinois.edu*
*University of Illinois Urbana-Champaign*

**David Pichler**                                                  *david.pichler@intelinair.com*
*Intelinair*

**Daniel Marley**                                                          *daniel@intelinair.com*
*Intelinair*

**David Wilson**                                                            *david@intelinair.com*
*Intelinair*

**Naira Hovakimyan**                                                        *nhovakim@illinois.edu*
*University of Illinois Urbana-Champaign*
*Intelinair*

**Jennifer Hobbs**                                                  *jenniferhobbs08@gmail.com*
*Intelinair*

**Reviewed on OpenReview:** *https://openreview.net/forum?id=v5jwDLqfQo*

## Abstract

A key challenge for much of the machine learning work on remote sensing and earth observation data is the difficulty in acquiring large amounts of accurately labeled data. This is particularly true for semantic segmentation tasks, which are much less common in the remote sensing domain because of the incredible difficulty in collecting precise, accurate, pixel-level annotations at scale. Recent efforts have addressed these challenges both through the creation of supervised datasets as well as the application of self-supervised methods. We continue these efforts on both fronts. First, we generate and release an improved version of the Agriculture-Vision dataset (Chiu et al., 2020b) to include raw, full-field imagery for greater experimental flexibility. Second, we extend this dataset with the release of 3600 large, high-resolution (10cm/pixel), full-field, red-green-blue and near-infrared images for pre-training. Third, we incorporate the Pixel-to-Propagation Module (Xie et al., 2021b) originally built on the SimCLR framework into the framework of MoCo-V2 (Chen et al., 2020b). Finally, we demonstrate the usefulness of this data by benchmarking different contrastive learning approaches on both downstream classifications *and* semantic segmentation tasks. We explore both CNN and Swin Transformer (Liu et al., 2021a) architectures within different frameworks based on MoCo-V2. Together, these approaches enable us to better detect key agricultural patterns of interest across a field from aerial imagery so that farmers may be alerted to problematic areas in a timely fashion to inform their management decisions. Furthermore, the release of these datasets will support numerous avenues of research for computer vision in remote sensing for agriculture.

## 1 Introduction

Massive annotated datasets like ImageNet have fostered the development of powerful and robust deeplearning models for natural images (Deng et al., 2009; He et al., 2016; Simonyan & Zisserman, 2014;

Krizhevsky et al., 2012; Russakovsky et al., 2015). However, creating large complex datasets is costly, time-consuming, and may be infeasible in some domains or for certain tasks. Simultaneously, vast amounts of unlabeled data exist in most domains. Among different self-supervised learning methods(Chang et al., 2022; Khan et al., 2022; Ziegler & Asano, 2022; Albelwi, 2022; Singh et al., 2018; Ziegler & Asano, 2022), Contrastive learning has recently emerged as an encouraging candidate for solving the need for large labeled datasets (Grill et al., 2020; Caron et al., 2020; 2018; He et al., 2020). Through pre-training, these approaches open up the possibility of using unlabeled images as its own supervision and transferring in-domain images to further downstream tasks (Tian et al., 2020; Ayush et al., 2020).

While natural scene imagery largely dominates the research landscape in terms of vision algorithms, datasets and benchmarks, the rapid increase in quantity and quality of remote sensing imagery has led to significant advances in this domain as well (Kelcey & Lucieer, 2012; Maggiori et al., 2017; Ramanath et al., 2019; Xia et al., 2017). Coupled with deep neural networks, remote sensing has achieved exceptional success in multiple domains such as natural hazards assessment (Van Westen, 2013), climate tracking (Rolnick et al., 2019; Yang et al., 2013), and precision agriculture (Mulla, 2013; Seelan et al., 2003; Wu et al., 2022; Barrientos et al., 2011; Gitelson et al., 2002). However, obtaining large quantities of accurate annotations is especially challenging for remote sensing tasks, particularly for agriculture, as objects of interest tend to be very small, high in number (perhaps thousands per image), possess complex organic boundaries, and may require channels beyond red-green-blue (RGB) to identify.

Approaches developed initially for natural images may work well on remote sensing imagery with only minimal modification. However, this is not guaranteed due to the large domain gap. Additionally, initial methods may fail to exploit the unique structure of earth observation data, such as geographic consistency or seasonality (Mañas et al., 2021). Explicitly benchmarking approaches on domain-relevant data is critical. In this work, we focus primarily on the Agriculture-Vision (AV) dataset (Chiu et al., 2020b): a large, multi-spectral, high-resolution (10 cm/pixel), labeled remote sensing dataset for semantic segmentation. Unlike low-resolution public satellite data, this imagery enables within-field identification of key agronomic patterns such as weeds and nutrient deficiency. While this dataset is noted for its size, most aerial agriculture datasets are quite small. Therefore we leverage the large amounts of *un-annotated data* which is readily available in this domain, benchmark several self-supervised approaches whose inductive bias reflects the structure of this data, and evaluate the impact of these approaches in more data-limited settings.

Together, our contributions are as follows:

- We release a full-field version of the Agriculture-Vision dataset to further encourage broad agricultural research in pattern analysis.
- We release over 3 terabytes of unlabeled, full-field images from more than 3600 full-field images to enable unsupervised pre-training.
- We benchmark self-supervised pre-training methods based on momentum contrastive learning and evaluate their performance on downstream classification *and* semantic segmentation tasks with variable amounts of annotated data.
- We perform benchmarks using both CNN and Swin Transformer backbones.
- We incorporate the Pixel-to-Propagation Module (Xie et al., 2021b) (PPM), originally built on SimCLR (Chen et al., 2020a), into the MoCo-V2 (Chen et al., 2020b) framework and evaluate its performance.
- We adapt the approach of Seasonal Contrast (SeCo) from Mañas et al. (2021) for this dataset, which contains imagery only during the growing season to address the spatiotemporal nature of the raw data specifically.

## 2 Related Work

### 2.1 Constrastive Learning

Unsupervised and self-supervised learning (SSL) methods have proven to be very successful for pre-training deep neural networks (Erhan et al., 2010; Bengio, 2012; Mikolov et al., 2013; Devlin et al., 2018). Recently, methods like MoCo (He et al., 2020; Chen et al., 2020b), SimCLR (Chen et al., 2020a), BYOL (Grill et al.,

2020) and others such as Bachman et al. (2019); Henaff (2020); Li et al. (2020) based on contrastive learning methods have achieved state-of-the-art performance. These approaches seek to learn by encouraging the attraction of different views of the same image ("positive pairs") as distinguished from "negative pairs" from different images (Hadsell et al., 2006). Several approaches have sought to build on these base frameworks by making modifications that better incorporate the invariant properties and structure of the input data or task output. Specifically pertinent to the current work, Xie et al. (2021b) extended the SimCLR framework through the incorporation of a pixel-to-propagation module and additional pixel-level losses to improve performance on downstream tasks requiring dense pixel predictions. Mañas et al. (2021) combined multiple encoders to capture the time and position invariance in downstream remote sensing tasks.

## 2.2 Remote Sensing Datasets

Aerial images have been widely explored over the past few decades (Cordts et al., 2016; Everingham et al., 2010; Gupta et al., 2019; Lin et al., 2014; Zhou et al., 2017), but the datasets for image segmentation typically focus on routine, ordinary objects or street scenes (Deng et al., 2009). Many prominent datasets including Inria Aerial Image (Maggiori et al., 2017), EuroSAT (Helber et al., 2019), and DeepGlobe Building (Demir et al., 2018) are built on low-resolution satellite (e.g. Sentinel-1, Sentinel-2, MODIS, Landsat) and only have limited resolutions that vary from 800 cm/pixel to 30 cm/pixel and can scale up to 5000×5000 pixels. Those datasets featuring segmentation tend to explore land-cover classification or change detection (Daudt et al., 2018; Sumbul et al., 2019).

Pertaining to aerial agricultural imagery, datasets tend to be either low-resolution (>10 m/pixel) satellite (Tseng et al., 2021; Feng & Bai, 2019) or very high-resolution (<1 cm/pixel) imagery taken from UAV or on-board farming equipment (Haug & Ostermann, 2014; Olsen et al., 2019). The Agriculture-Vision dataset (Chiu et al., 2020b;a) introduced a large, high-resolution (10 cm/pixel) dataset for segmentation, bridging these two alternate paradigms.

# 3 Datasets

## 3.1 Review and Reprocessing of Agriculture-Vision Dataset

The original AV dataset (Chiu et al., 2020b) consists of 94,986 labeled high-resolution (10-20 cm/pixel) RGB and near-infrared (NIR) aerial imagery of farmland. Special cameras were mounted to fixed-wing aircraft and flown over the Midwestern United States during the 2017-2019 growing seasons, capturing predominantly corn and soybean fields. Each field was annotated for nine patterns described in the supplemental material. After annotation, $512 \times 512$ *tiles* were extracted from the full-field images and *then* pre-processed and scaled. While this pre-processing produces a uniformly curated dataset, it naturally discards important information about the original data.

To overcome this limitation, we obtained the original raw, full-field imagery. We are releasing this raw data as full-field images without any tiling, as it has been demonstrated to be beneficial to model performance (Chiu et al., 2020a). A sample image is shown in Figure 1 (left). The original dataset can be recreated from this new dataset by extracting the tiles at the appropriate pixel coordinates provided in the data manifest.

## 3.2 Raw Data for Pre-training

We identified 1200 fields from the 2019-2020 growing seasons collected in the same manner as in Section 3.1. For each field, we selected three images, referred to as *flights*, taken at different times in the growing season, resulting in 3600 raw images available for pre-training. We elect to include data from 2020 even though it is not a part of the original supervised dataset because it is of high quality, similar in distribution to 2019, and we wish to encourage exploration around incorporating different source domains into modeling approaches as this is a very central problem to remote sensing data. We denote this raw imagery plus the original supervised dataset (in full-field format) as the "Extended Agriculture-Vision Dataset" (AV+); it will be made publicly available. The statistics of AV+ compared with AV are demonstrated in Table 1.

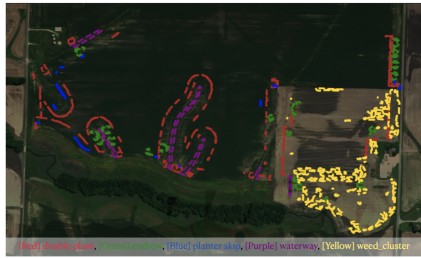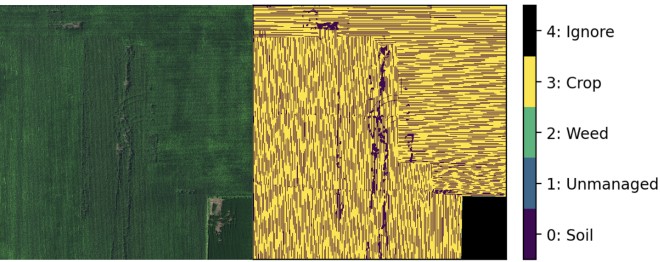

Figure 1: Left: Full-field imagery (RGB-only) constructed from the AV dataset. A field of this size is approximately 15,000×15,000 pixels which can yield many smaller tiles. Right: Sample imagery and labels for the fine-grained segmentation task.

One characteristic of remote sensing is data revisiting: capturing images from the same locations multiple times. Through data revisiting, the temporal information can serve as an additional dimension of variation beyond the spatial information alone. In AV+, a typical revisit time ranges from seven days to six months, capturing a field at different points during the pre-planting, growing, and harvest seasons. We provide an example of a revisit in Figure 3.

Table 1: Statistics of Agriculture Vision(Chiu et al., 2020b) and Extended Agriculture Vision(the part of raw imagery). We provide information about the number of images, image size, pixel numbers, color channels and the ground sample resolution (GSD). "cls.", "seg." and "SSL" stand for classification, segmentation and self-supervised learning respectively.

| Dataset | # of Images | Tasks | Image Size | Channels | Resolution (GSD) | # of pixels |
|---|---|---|---|---|---|---|
| AV | 94,986 | Cls./Seg. | 512 × 512 | RGB, NIR | 10/15/20 cm/px | 22.6B |
| AV+ | 3600 | SSL | 15,000 × 15,000 | RGB, NIR | 10/15/20 cm/px | 810.0B |

### 3.3 Fine-Grained Segmentation Dataset

Fine-grained segmentation tasks for high-resolution remote sensing data, particularly for agriculture, are often overlooked because of the difficulty in collecting sufficient amounts of annotated data(Monteiro & von Wangenheim, 2019; Haug & Ostermann, 2015). To explore the transferability of the AV+ dataset and SSL methods to a very challenging, data-limited, in-domain (i.e. same sensor and geography) task, we construct a densely annotated dataset. We collected 68 flights from the 2020 growing season that were not included in AV+ for this task. From these flights, 184 tiles with shape 1500×1500 were selected and densely annotated with four classes: soil, weeds, crops, and un-managed area (e.g. roads, trees, waterways, buildings); an "ignore" label was used to exclude pixels which may unidentifiable due to image collection issues, shadows, or clouds. The annotations in this dataset are much more fine-grained than those in the AV+ dataset. For example, whereas the AV+ dataset identifies regions of high weed density as a "weed cluster", this dataset identifies each weed individually at the pixel level and also labels any crop or soil in those regions by their appropriate class. A sample image and annotation are shown in Figure 1 (right).

The fine-grained nature and small dataset size make this a very challenging segmentation task: very young crops often look like weeds, weeds growing among mature crops are only detectable through an interruption in the larger structure of the crop row, unmanaged areas often contain grasses and other biomass which closely resemble weeds but is not of concern to the grower, and classes are highly imbalanced.

## 4 Methodology for Benchmarks

In this section, we present multiple methods for pre-training a transferable representation on the AV+ dataset. These methods include MoCo-V2 (Chen et al., 2020b), MoCo-V2 with a Pixel-to-Propagation Module (PPM) (Xie et al., 2021b), the multi-head Temporal Contrast based on SeCo (Mañas et al., 2021), and

a combined Temporal Contrast model with PPM. We also explore different backbones based on ResNet (He et al., 2016) and the Swin Transformer architecture (Liu et al., 2021a).

## 4.1 Momentum Contrast

MoCo-V2 is employed as the baseline module for the pre-training task. Unlike previous work focusing only on RGB channels (He et al., 2020; Mañas et al., 2021; Chen et al., 2020b), we include the information and learn representations from RGB and NIR channels. In each training step of MoCo, a given training example $x$ is augmented into two separate views, query $x^q$ and key $x^k$. An online network and a momentum-updated offline network, map these two views into close embedding spaces $q = f_q(x^q)$ and $k^+ = f_k(x^k)$ accordingly; the query $q$ should be far from the negative keys $k^-$ coming from a random subset of data samples different from $x$. Therefore, MoCo can be formulated as a form of dictionary lookup in which $k+$ and $k-$ are the positive and negative keys. We define the instance-level loss $\mathcal{L}_{inst}$ with temperature parameter $\tau$ for scaling (Wu et al., 2018) and optimize the dictionary lookup with InfoNCE (Oord et al., 2018):

$$\mathcal{L}_{inst} = -\log \frac{\exp(q \cdot k^+/\tau)}{\sum_{k^-} \exp(q \cdot k^-/\tau) + \exp(q \cdot k^+/\tau)} \tag{1}$$

## 4.2 Momentum Contrast with Pixel-to-Propagation Module

Compared with classical datasets such as ImageNet (Deng et al., 2009), COCO (Lin et al., 2014), and LVIS (Gupta et al., 2019) in the machine learning community, low-level semantic information from AV+ is more abundant, with regions of interest corresponding more closely to "patterns" (i.e. areas of weed clusters, nutrient deficiency, storm damage) and less to individual instances. Therefore, pre-training MoCo-V2 beyond image-level contrast should be beneficial to downstream pattern analysis tasks.

### 4.2.1 Pixel-to-Propagation Module

Xie et al. (2021b) added a Pixel-Propagation-Module (PPM) to the SimCLR framework and achieved outstanding results on dense downstream tasks. In PPM, the feature of a pixel $x_i$ is smoothed to $q_i^s$ by feature propagation with all pixels $x_{\hat{i}}$ within the same image $I$ following the equation:

$$q_i^s = \sum_{x_j \in I} S(x_i, x_{\hat{i}}) \cdot G(x_{\hat{i}}), \tag{2}$$

where $G(\cdot)$ is a transformation function instantiated by linear and ReLU layers. $S(\cdot, \cdot)$ is a similarity function defined as

$$S(x_i, x_{\hat{i}}) = (max(cos(x_i, x_{\hat{i}}), 0))^\gamma \tag{3}$$

with a hyper-parameter $\gamma$ to control the sharpness of the function.

### 4.2.2 Extend Pixel-to-Propagation Module to MoCo

Notably, previous work (Xie et al., 2021b) based on SimCLR requires a large batch size, which is not always achievable. To generalize the PPM and make the overall pre-training model efficient, we incorporate the pixel-level pretext tasks into basic MoCo-V2 models to learn dense feature representations. As demonstrated in Figure 2, we add two extra projectors for pixel-level pretask compared with MoCo-V2. The features from the backbones are kept as feature maps instead of vectors to ensure pixel-level contrast. To decide positive pairs of pixels for contrast, each feature map is first warped to the original image space. Then the distances between the pixel $i$ and pixel $j$ from each of the two feature maps are computed and normalized. Given a hyper-parameter $\tau$ (set as 0.7 by default), $i$ and $j$ are recognized as one positive pair if their distance is less than $\tau$. Then, we can compute the similarity between two pixel-level feature vectors, i.e., smoothed $q_i^s$ from PPM and $k_j$ from the feature map for each positive pair of pixels $i$ and $j$. Since two augmentation views both pass the two encoders, we use a loss in a symmetric form following Xie et al. (2021b):

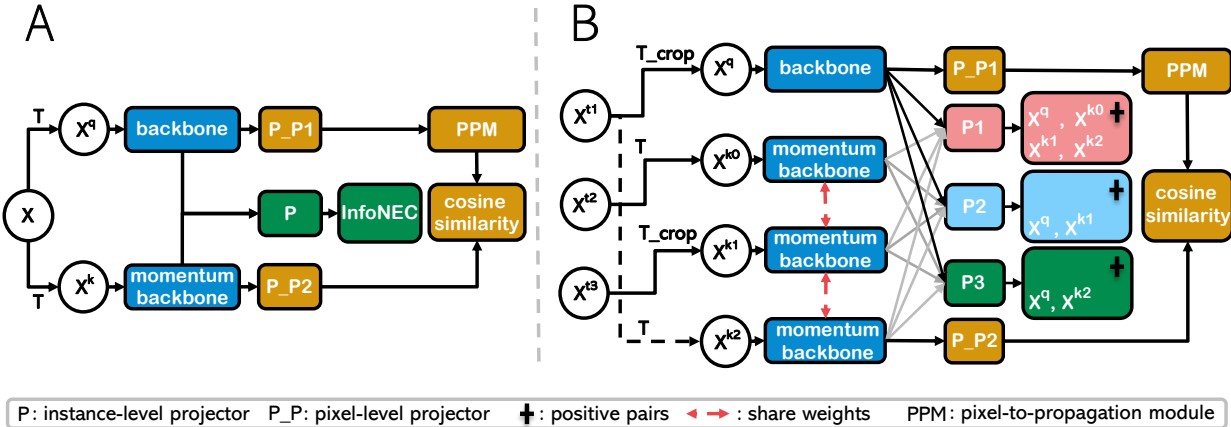

Figure 2: **A**. Diagram of MoCo-V2 with Pixel-to-Propagation Module (MoCo-PixPro). $P$ includes a normally updated projector and a momentum-updated projector. For pixel-level pre-task, $P\_P1$ is updated by gradient descent and $P\_P2$ is the momentum projector. **B**. Diagram of Temporal Contrast with Pixel-to-Propagation Module (TemCo-PixPro). Query view $x^q$ and key view $x^{k_0}$ contain both artificial and temporal variance. Query view $x^q$ and key view $x^{k_1}$ contain only temporal variance. Query view $x^q$ and key view $x^{k_2}$ only contain artificial variance. Identical cropping $T_{crop}$ is applied to $x^{t_1}$ and $x^{t_3}$. Pixel-level contrast in only computed on $x^q$ and $x^{k_2}$. For these two sub-plots, modules in navy blue ■ serve as encoders for feature extraction. Modules in brown ■ are designed for pixel contrast, which includes projectors, pixel propagation modules and the loss being used. Pink modules ■ represent instance-level contrast with embeddings space invariant to all kinds of augmentations. Similarly, modules in green ■ and sky blue ■ mean instance-level contrast but extract features invariant to artificial augmentation and temporal augmentation, respectively.

$$\mathcal{L}_{PixPro} = -cos(q_i^s, k_j) - cos(q_j^s, k_i) \tag{4}$$

During the training, the loss $\mathcal{L}_{PixPro}$ from the PPM is integrated with the instance-level loss as shown in the equation 5. These two complementary losses are balanced by a factor $\alpha$, set to 0.4 in all the experiments (see Supplemental: Additional Results - Balance Factor).

$$\mathcal{L} = \alpha\mathcal{L}_{inst} + \mathcal{L}_{PixPro} \tag{5}$$

### 4.3 Temporal Contrast

While a pixel-level pretext task learns representations useful for spatial inference, we would like to learn a representation that takes advantage of the temporal information structure of AV+. Following the work of SeCo (Mañas et al., 2021), additional embedding sub-spaces that are invariant to time are created. Since the backbones learn temporal-aware features through extra sub-spaces, it offers a more precise and general pattern analysis in downstream tasks. More specifically, we define a positive temporal pair by obtaining one pair of images from the same area (i.e. of the same field) but at different times as shown in Figure 3. We explore whether the structure provided by the temporal alignment of positive temporal pairs provides more semantically significant content than naive artificial transformations (i.e. flipping, shifting) for pre-training.

Unlike in SeCo where images were separated with a constant time (3 months), the time difference between images from our data varies from 1 week to 6 months. We adapt SeCo as follows. First, we randomly select three tiles with $512 \times 512$ from the same field at identical locations but different times, which will be defined as $x^{t_1}$, $x^{t_2}$ and $x^{t_3}$. Only random cropping $T_{crop}$ is applied to the query image to generate the query view, i.e., $x^q = T_{crop}(x^{t_1})$. The first key view that contains both temporal and artificial variance is defined as $x^{k_0}$

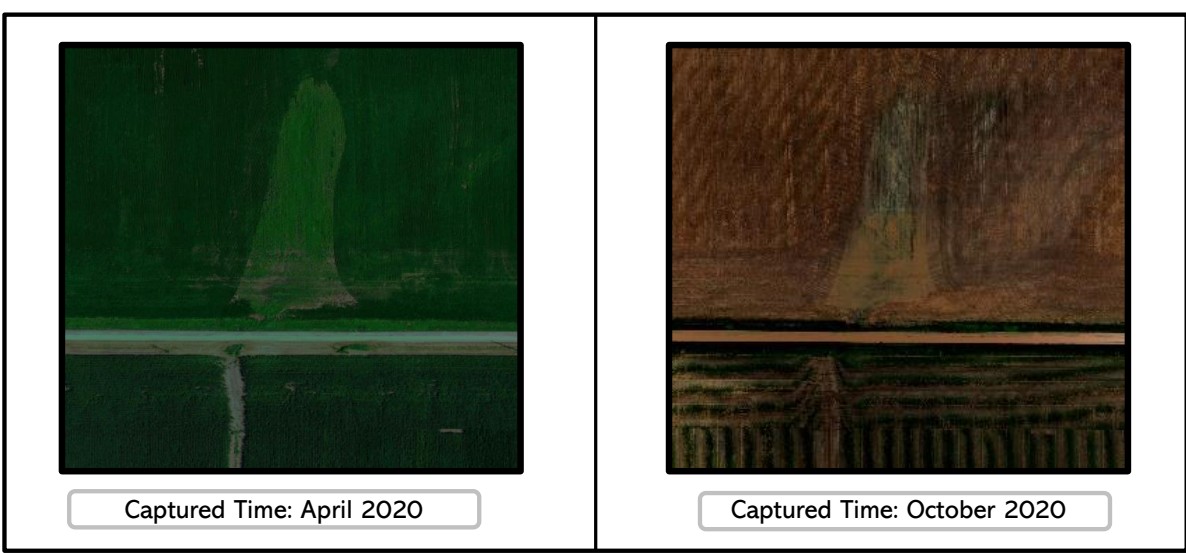

Figure 3: Visualization of temporal contrast in AV+.

$= T(x^{t_1})$, where the $T$ is the typical data augmentation pipeline used in MoCo. The second key contains only temporal augmentation compared with the query view. Therefore, we apply the exact same cropping window applied to the query image, $x^{k_1} = T_{crop}(x^{t_2})$. The third key contains only artificial augmentations, $x^{k_2} = T(x^{t_0})$. Following the MoCo and SeCo learning strategy (He et al., 2020; Mañas et al., 2021), these views can be mapped into three sub-spaces that are invariant to temporal augmentation, artificial augmentation and both variance. In this way, we fully explore the multi-time scale information in AV+ to improve the temporal sensitivity of encoders further. Since the temporal contrast does not necessarily cross seasons or enforce alignment of seasonality within a sub-space, we denote our approach as Temporal Contrast (TemCo).

### 4.4 Temporal Contrast with Pixel-to-Propagation Module

We create an integrated model (TemCo-PixPro) to capture the dense, spatiotemporal structure of AV+. Concretely, we merge PPM and TemCo into a single model to increase the encoders' spatial and temporal sensitivity.

To ensure efficient computation, we do not compute pixel-wise contrastive updates in each temporal sub-space. Instead, we assign two extra projectors for pixel-level contrastive learning. We include the PPM after the online backbone and one of the pixel-level projectors to smooth learned features. Then, we calculate the similarity of the smooth feature vectors and the momentum encoder features through a dot product. We illustrate the overall architecture of this model in Figure 2B.

### 4.5 Swin Transformer-Based Momentum Contrast

While the Swin Transformer achieves superior performance on various computer vision tasks (Liu et al., 2021a;b), only very recent work has focused on self-supervised training for vision transformers (ViT)(Xie et al., 2021a; Li et al., 2021). To the best of our knowledge, no study has investigated Swin Transformer's performance on remote sensing datasets using self-supervised methods. Therefore, we explore a Swin Transformer-based MoCo for pre-training of AV+. Specifically, we adopt the tiny version of the Swin Transformer (Swin-T) as the default backbone.

Following most transformer-based learning tasks, we adopt AdamW (Kingma & Ba, 2014) for training. Additionally, we incorporate the multiple-head projectors from TemCo and PPM to capture temporal knowledge and pixel-level pretext tasks.

# 5 Experiments and Results

We benchmark the performance and transferability of the learned representations on four downstream tasks: agricultural pattern classifications, semantic segmentation on AV+, fine-grained semantic segmentation and land-cover classification.

We report the implementation details and results from four models covering two kinds of backbones: basic Momentum Contrast model (MoCo-V2), basic Momentum Contrast with Pixel-to-Propagation Module (MoCo-PixPro), Temporal Contrast (TemCo), Temporal Contrast with Pixel-to-Propagation Module (TemCo-PixPro), Swin Transformer-Based MoCo-V2, Swin Transformer-Based TemCo, and Swin Transformer-Based TemCo-PixPro.

## 5.1 Pre-training Settings

y**Dataset.** We take AV+ as the dataset to pre-train the backbones. To be specific, all the raw images are cropped tiles with a shape $512 \times 512$ without overlapping, following the settings in Chiu et al. (2020b). Therefore, we eventually have 3 million cropped images. For each tile, it will be fed to the contrastive learning pipelines.

**Implementation Details.** We train each model for 200 epochs with batch size 512. For ResNet-based models, we use SGD as the optimizer with a weight decay of 0.0001 and momentum of 0.9. The learning rate is set to 0.03 initially and is divided by 10 at epochs 120 and 160. Swin-T models use the AdamW optimizer, following previous work (Xie et al., 2021a; Liu et al., 2021a). The initial learning rate is 0.001, and the weight decay is 0.05. All the artificial data augmentations used in this paper follow the work of MoCo-V2 as this data augmentation pipeline archives the optimal performance. These data augmentations include random color jitter, gray-scale transform, Gaussian blur, horizontal flipping, resizing, and cropping.

We use all four channels, RGB and NIR, to fully extract the features contained in the dataset. When testing ImageNet-initialized backbones for comparison, we copy the weights corresponding to the Red channel of the pre-trained weights from ImageNet to the NIR channel for all the downstream tasks following the method of Chiu et al. (2020b).

## 5.2 Downstream Classifications

### 5.2.1 Preliminaries

**Protocols.** The downstream classification task considers nine patterns: Nutrient Deficiency, Storm Damage, Drydown, Endrow, Double plant and Weeds. More details descriptions of these patterns can be found in the supplementary. We benchmark and verify the performance of our basic model (MoCo-V2) and its variants on the classification task of the labeled portion of AV+, following three protocols: (i) linear probing (He et al., 2020; Chen et al., 2020b;a; Tian et al., 2020), (ii) non-linear probing (Han et al., 2020), and (iii) fine-tuning the entire network for the downstream task.

**Dataset.** For this classification task, there are a total of 94,986 tiles with a size of $512 \times 512$ from AV. With a 6/2/2 train/val/test ratio, AV contains 56,944/18,334/19,708 train/val/test images following the exact same settings from Chiu et al. (2020b). We train the classifiers on the training set and evaluate their performance in the validation set.

### 5.2.2 Linear Probing

Following standard protocol, we freeze the pre-trained backbone network and train only a linear head for the downstream task. We train the models for 50 epochs using Adam optimizer with an initial learning rate of 0.0001 and report the top-1 classification validation set.

Figure 4 shows the impact of different weight initialization and percentages of labeled data in the downstream task. Consistent with previous research (Mañas et al., 2021), there is a gap between remote sensing and natural image domains: ImageNet weights are not always an optimal choice in this domain. MoCo-PixPro

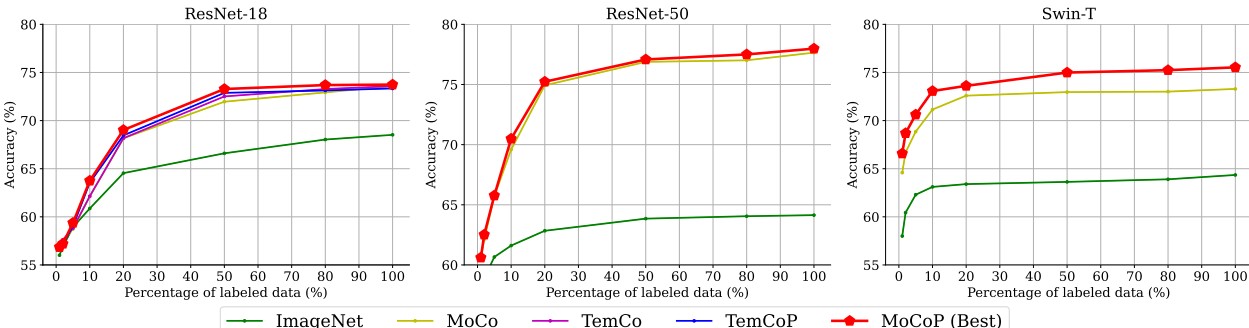

Figure 4: Accuracy under the linear probing protocol on AV+ classification. Results are shown from different pre-training approaches with different backbones.

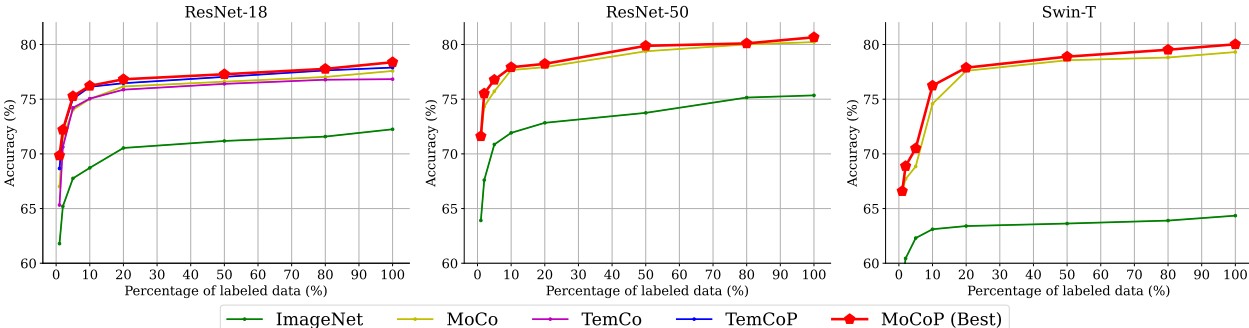

Figure 5: Accuracy under the non-linear probing protocol on AV+ classification. Results are shown from different pre-training approaches with different backbones, ResNet-18 (left), ResNet-50 (middle), and Swin-T (right), under different percentages of labeled data for the downstream task.

obtains the highest accuracy for the ResNet-18 backbone. As we compare the results of ResNet50 and Swin-T with fully labeled data, all Swin-T models underperformed their CNN counterparts.

### 5.2.3 Non-Linear Probing

We evaluate the frozen representations with non-linear probing: a multi-layer perceptron (MLP) head is trained as the classifier for 100 epochs with Adam optimization.

Classification results on AV+ classification under non-linear probing are shown in Figure 5. Consistent with results in the natural image domain (Han et al., 2020), non-linear probing results surpass linear probing. Our SSL weights exceed ImageNet's weights by over 5% regardless of the amount of downstream data or backbone type. From the results of ResNet-18, the optimal accuracy between different pre-training strategies comes from either MoCo-PixPro or TemCo-PixPro, which is different from linear probing. Overall, MoCo-PixPro performs better than the basic MoCo model across different backbones.

### 5.2.4 Fine-Tuning

Finally, we examine end-to-end fine-tuning with different percentages of labeled AV+ data for classification. We use the same architecture, learning schedule and optimizer as non-linear probing.

Our SSL weights show outstanding results in the low-data regions (<10% of data). For ResNet-18, MoCo-PixPro is better than the other models in all cases, whereas other SSL models demonstrate similar performance to ImageNet when labeled data is abundant. As we increase the backbone size to ResNet-50, our MoCo and MoCo-PixPro stably outperform ImageNet's model across all amounts of data, suggesting a greater capacity to learn domain-relevant features.

In Figure 6 (right), all models perform agreeably well in the Swin-T framework compared with weights from ImageNet. While fine-tuning was performed in the same manner as the ResNet models for fair comparisons, Swin-T shows the most promising performance in this end-to-end setting.

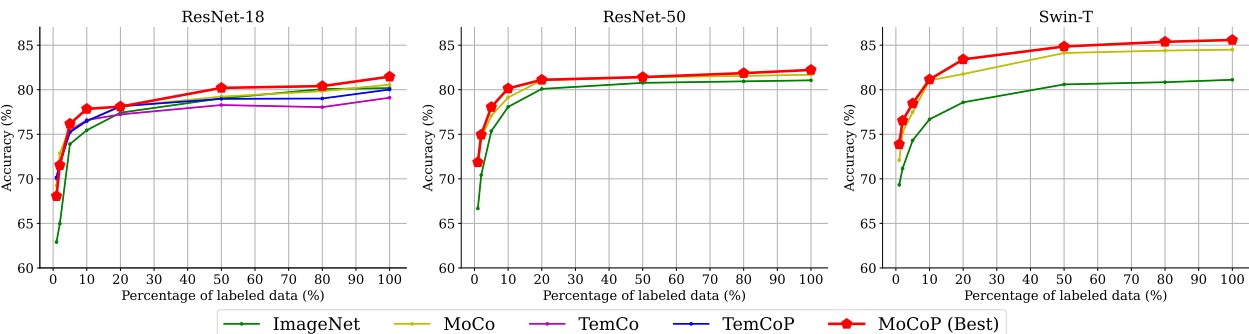

Figure 6: The accuracy under the end-to-end classification protocol on AV+. Results cover different pre-training approaches and backbones, varying from ResNet-18 (left), ResNet-50 (middle), and Swin-T (right). We also report the model's performance tuned with different percentages of the fully labeled dataset, ranging from one percent to a hundred percent.

### 5.3 Semantic Segmentation on Extended Agriculture-Vision

#### 5.3.1 Preliminaries

**Protocols.** We continue our benchmarking study by examining their impact on the semantic segmentation approach on AV+ as originally formulated in Chiu et al. (2020b). Again, we apply two protocols for evaluating the learned representations: maintaining a fixed encoder or fine-tuning the entire network.

To naively assess the pre-trained representations, we adopt the simple yet effective U-Net (Ronneberger et al., 2015) framework. Unlike previous work on AV (Chiu et al., 2020b), we report results over all the patterns in AV+, including storm damage, to ensure an integrated and comprehensive analysis.

First, we evaluate the representation by holding the pre-trained encoder fixed and fine-tuning only the decoder during the supervised learning phase. Similarly, we evaluate the pre-training impact on segmentation tasks allowing for fine-tuning of both the encoder and decoder during the supervised learning phase. We train the models using Adam optimization with an initial learning rate of 0.01. The one-cycle policy (Smith, 2017) is used to update the learning rate as in Chiu et al. (2020b). ResNet-18 models are trained for 30,000 steps, while the larger ResNet-50 and Swin-T models are trained for 120,000 steps to allow for sufficient training.

**Dataset.** For this segmentation task, there are a total of 94,986 tiles with a size of $512 \times 512$. With the same 6/2/2 train/val/test ratio, AV contains 56,944/18,334/19,708 train/val/test images following settings from Chiu et al. (2020b) and the previous classification task. We train the classifiers on the training set and evaluate their performance in the validation set.

#### 5.3.2 Segmentation Results

As results are shown in Table 2, MoCo-PixPro performs the best for the ResNet-18 backbone when the encoder remains fixed during supervised training; this result is similar to that seen for classification. This result supports our hypothesis that AV+ has abundant low-level semantic information and including pixel-level pre-task is critical for downstream learning tasks. When the encoder is unfrozen during supervised training, the basic MoCo-V2 shows the best results, but is not significantly better than TemCo or TemCo-PixPro. By scaling from ResNet-18 to ResNet-50, MoCo-PixPro outperforms ImageNet, especially when the encoder remains fixed. Importantly, unlike the ResNet-based models, the Swin Transformer-based MoCo-PixPro shows the best results across all variations in the setting. Another important observation is that the PPM benefits more as we scale up the models from ResNet-18 to ResNet-50 and then Swin-T. As the training epochs are all the same for all the pre-training, smaller backbones like ResNet-18 are more likely to get overfitted. When trained with ResNet-50, the performance drop of MoCo-PixPro is very small compared with MoCo. As we move to Swin-T, MoCo-PixPro eventually shows the best performance over other methods.

Table 2: Results of Downstream Segmentation Task on AV+ using mean-IoU metric

| Pretrained Weights | Backbone | mIoU (%) Fixed 1% | mIoU (%) Fixed 100% | mIoU (%) Fine-Tuned 1% | mIoU (%) Fine-Tuned 100% |
|---|---|---|---|---|---|
| Random | ResNet-18 | 18.89 | 21.37 | 19.02 | 26.94 |
| ImageNet | ResNet-18 | 19.02 | 23.39 | 19.73 | 29.23 |
| MoCo-V2 | ResNet-18 | 22.36 | 27.83 | **22.53** | **31.80** |
| MoCo-PixPro | ResNet-18 | **23.71** | **30.60** | 20.04 | 30.56 |
| TemCo | ResNet-18 | 23.71 | 26.85 | 21.09 | 31.76 |
| TemCo-PixPro | ResNet-18 | 22.97 | 28.60 | 21.32 | 31.66 |
| Random | ResNet-50 | 19.42 | 21.82 | 18.71 | 26.37 |
| ImageNet | ResNet-50 | 21.21 | 25.94 | 20.31 | 30.52 |
| MoCo-V2 | ResNet-50 | 24.25 | 31.03 | **21.47** | **31.87** |
| MoCo-PixPro | ResNet-50 | **25.76** | **32.35** | 21.36 | 31.58 |
| Random | Swin-T | 15.89 | 20.10 | 22.68 | 37.14 |
| ImageNet | Swin-T | 20.00 | 22.40 | 30.96 | 43.01 |
| MoCo-V2 | Swin-T | 25.51 | 30.60 | 28.12 | 41.02 |
| MoCo-PixPro | Swin-T | **27.61** | **32.96** | **32.06** | **43.33** |

### 5.3.3 Comparison with Agriculture-Vision Results

The AV dataset was benchmarked on a downstream segmentation task with architectures based on the DeepLabV3 (Chen et al., 2018) framework. Since the previous results report mean Intersecion-over-Union (mIoU) for 8 agricultural patterns, we re-trained our models using a U-Net architecture (Ronneberger et al., 2015) including on more pattern, i.e., the storm damage. With a lightweight U-Net, smaller backbone, and much less training, our SwinT-based model outperforms the best results from Chiu et al. (2020b) in the Table 3, demonstrating the effectiveness of our approach. Additionally, we demonstrate the effectiveness of pre-training and fine-tuning this multi-spectral data. AV and AV+ are beyond most conventional images, consisting of NIR-Red-Green-Blue (NRGB) channels. Therefore, we investigate the differences in semantic segmentation performance from multi-spectral images, including regular RGB and RGBN images. According to Table 3, NIR channels benefit the segmentation results over different backbones and segmentation methods.

For this eight-class segmentation task, we train the nine-class models using an Adam optimization with an initial learning rate of 0.01 and the one-cycle policy (Smith, 2017) for the learning rate adjustment. For fair comparisons and to be consistent with the supervised learning settings in Chiu et al. (2020b), we use a batch size of 40 and 25,000 iterations with warmup training for 1,000 iterations.

Table 3: Comparison of mIoUs between the Agriculture-Vision model and our proposed U-Net-based model on Agriculture-Vision validation set.

| Methods | Pre-trained Weights | Backbone | Channels | mIOU(%) | # Parameters |
|---|---|---|---|---|---|
| FPN(Chiu et al., 2020b) | ImageNet | ResNet-101 | RGB | 40.48 | 45.10M |
| U-Net | MoCo-V2 | Swin-T | RGB | 44.77 | 32.40M |
| U-Net | MoCo-PixPro | Swin-T | RGB | **45.92** | 32.40M |
| FPN(Chiu et al., 2020b) | ImageNet | ResNet-101 | RGBN | 43.40 | 45.11M |
| U-Net | MoCo-V2 | Swin-T | RGBN | 46.15 | 32.40M |
| U-Net | MoCo-PixPro | Swin-T | RGBN | **48.75** | 32.40M |

Table 4: IoU for each model in the fine-grained semantic segmentation task considering different encoder weight initialization, architectures, and weight fixing schemes.

| Weights | Architecture | IoU (Fixed-Weights) | IoU (Fine-Tuned) |
|---|---|---|---|
| Random | ResNet-18 | 39.05 | 42.19 |
| ImageNet | ResNet-18 | 40.81 | **45.47** |
| MoCo-V2 | ResNet-18 | **44.05** | 43.97 |
| MoCo-PixPro | ResNet-18 | 42.03 | 44.62 |
| TemCo | ResNet-18 | 42.30 | 44.48 |
| TemCo-PixPro | ResNet-18 | 43.45 | 43.91 |
| MoCo-V2 | ResNet-50 | 40.03 | 40.54 |
| MoCo-V2 | Swin-T | 40.25 | 40.00 |
| MoCo-PixPro | Swin-T | 37.56 | 40.67 |
| TemCo. | Swin-T | 39.52 | 40.26 |

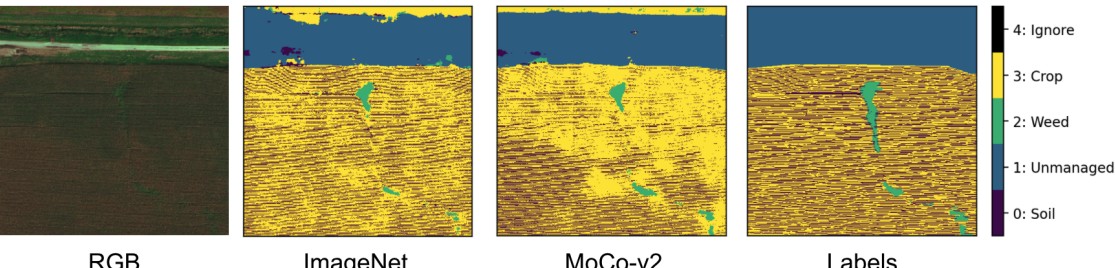

Figure 7: A sample output on the fine-grained segmentation task using fixed-encoder weights from ImageNet and MoCo-V2. The segmentation outputs are compared with both the original RGB image and the segmentation labels.

## 5.4 Fine-Grained Semantic Segmentation

Unlike AV+, this dataset is severely limited by the availability of fine-grained segmentation labels. There are 184 tiles, from 68 flights, in this dataset that are split into training (70%), validation (15%), and test (15%). Again, we use a U-Net architecture with a ResNet-18 encoder. For training, we use a multi-class focal loss Lin et al. (2017) to account for the strong class imbalance.

Results are shown in Table 4, and sample output is shown in Figure 7. Results improve across the board when both the encoder and decoder are fine-tuned. Although less dramatic than the results seen on the AV+ classification and segmentation tasks, some improvement over ImageNet weights is seen using the MoCo-v2 framework with ResNet-18 backbone for fixed weights. As seen on the other tasks, when the entire network undergoes fine-tuning, the ImageNet and SSL weights, specifically MoCo-PixPro, produce roughly the same performance on the downstream task. Additional per-class analysis is provided in the Supplemental. The ResNet-50 and Swin-T models performed relatively worse compared to the ResNet-18 models, which is unsurprising given the extremely small size of this dataset.

## 5.5 Land-Cover Classification on EuroSAT

We further prove pretraining on the AV+ dataset benefits the downstream task in the broader remote sensing community. We conduct downstream classification experiments on EuroSAT (Helber et al., 2019). EuroSAT addresses the classification challenge of land use and land cover with images from Sentinel-2. It consists of 27,000 labeled images and 10 classes over 34 European countries. We use the splits protocol of train/val following the work of Neumann et al. (2019).

We freeze the pre-trained backbones and add a linear layer to evaluate the learned representation in this classification task. Totally, the linear layer is tunned with 100 epochs using the Adam optimizer. The initial learning rate is set to 0.001 and is divided by 10 at the 60th and 80th epochs.

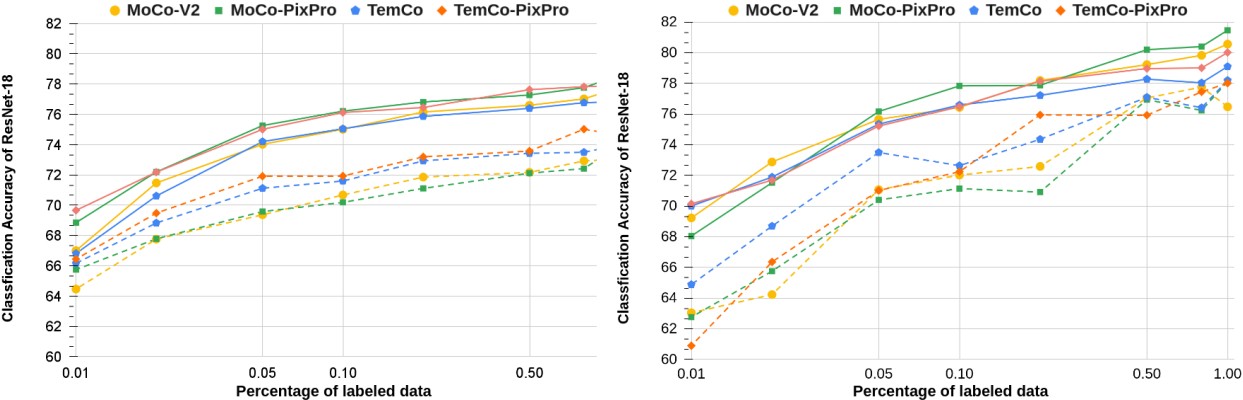

Figure 8: Ablation study on the pre-training size of data on different pre-training methods on two downstream tasks. Solid lines represent accuracy from 3600 flights while dashed lines represent accuracy from 1200 flights. Left: results from non-linear probing on downstream classification. Right: fine-tuning results on entire networks for downstream AV+ segmentation.

The results shown in the Table 5 compare weights pre-trained from AV+ against other baselines. We notice that MoCo-V2 and our proposed MoCo-PixPro achieve 1.21% and 6.25% higher accuracy compared with ImageNet's weights accordingly. These results confirm not only the effectiveness of pre-training on AV+ but also AV+'s significant potential to generalize to the broader remote sensing field.

Table 5: Accuracy of the EuroSAT land-cover classification task using ResNet-18

| Weights | Random | ImageNet | MoCo-V2 | MoCo-PixPro |
|---|---|---|---|---|
| Accuracy (%) | 63.21 | 86.32 | 87.53 | **89.97** |

### 5.6 Ablation Study: Number of Flights

We use a ResNet-18 backbone and basic MoCo-V2 for experiments. When the number of flights used for SSL is increased from 300 to 3600, we observe stable improvement in the downstream classification task under the non-linear probing setting; this gain is confirmed regardless of the fraction of the labeled dataset for tuning. See Supplemental: Additional Results for more detailed results.

This improvement is seen for all examined SSL methods Figure 8 when the raw dataset is increased from 1200 to 3600 flights and evaluated under non-linear probing for classification and full-network fine-tuning for AV+ segmentation. Our SSL models' performance steadily grows as raw data size increases, suggesting that even more data may lead to even greater performance.

## 6 Conclusion

Large, high-quality datasets are opening tremendous new opportunities for computational agriculture, but they are extremely difficult to obtain. As in other domains, remote sensing and earth observation data are marked by huge amounts of unlabeled data and relatively few annotations; leveraging the information in this unlabeled data, therefore, becomes a critical task. In this work, we contribute to the advancement of these efforts by releasing the AV+ dataset, which contains annotated full-field imagery based on the original AV dataset Chiu et al. (2020b), supplemented by more than 3TB of raw full-field images taken at different times in the season. The improved supervised component of the AV dataset will allow for greater flexibility in training and augmentation protocols and enable additional possible lines of study around long-range context and large-scale imagery. The raw unlabeled data will enable continued exploration in the self, semi, and weakly supervised methods which we have begun to benchmark here. This extension of an already important

dataset in computational agriculture will open up many lines of research and investigation which benefit both the agriculture and computer vision communities.

Next, we conduct a thorough benchmark study on self-supervised pre-training methods based on contrastive learning, which captures the fine-grained, spatiotemporal nature of this data. We analyze a classification formulation of the AV+ dataset under linear probing, non-linear probing, and fine-tuning. We also examined segmentation tasks, which are often overlooked in remote sensing approaches, based on the original segmentation formulation of AV+ with a frozen and unfrozen encoder and an extremely small fine-grained segmentation task under the same formulations. Our benchmark study explores both traditional CNN architectures (ResNet-18 and ResNet-50) as well as the more recent Swin Transformer, which offers unique potentials for computer vision, but requires huge amounts of data to train.

Importantly, we incorporate the Pixel-to-Propagation Module, originally built in the SimCLR framework, into the MoCo-V2 framework, which allows for training on larger batch sizes. Our results show that this module is key for downstream segmentation and *classification* tasks, even though it was designed primarily for dense detection and segmentation tasks. As our dataset contains richer low-level, high-frequency, fine-grained features than traditional natural imagery like COCO or ImageNet, this suggests that PPM is beneficial for learning dense, fine-grained *features* in addition to dense label structure.

We further combine this module with a TemCo, a modification of SeCo, into a rich framework that captures the dense, spatiotemporal structure of our data. While this combined framework was not the highest-performing on the various task, it again may have been at a disadvantage since it is a larger model and the number of steps was fixed for a fair comparison. Additionally, extending how *positive* samples are generated could prove beneficial. These improvements are the focus of future analysis.

Self-supervised methods will be crucial for unlocking opportunities in remote sensing, particularly for agriculture, and this dataset release and benchmark study offer a significant step in that direction.

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
