# OpenReview forum: "Extended Agriculture-Vision: An Extension of a Large Aerial Image Dataset for Agricultural Pattern Analysis"
_TMLR — Accepted by TMLR_

### Review · Reviewer_GJ9C · 2022-12-05

**Summary Of Contributions:**

This paper proposes a new release that extends the Agriculture-Vision dataset. Specifically, the proposed version contains more than 3,600 full-field images to encourage self-supervised representation learning. Further, this paper benchmarks various self-supervised methods on this new dataset with extensive studies using different backbones (e.g., ResNet and SwinT) and contrastive learning methods (e.g., Pixel-to-Propagation, MoCo-V2, and Seasonal Contrast). Finally, this paper adapts the SeCo to handle the dense and spatiotemporal structure of the image data captured over time.

**Audience:**

Yes

**Broader Impact Concerns:**

I am concerned if researchers in general remote sensing will be interested in using the dataset, as the domain-specific knowledge has not been explained well in the manuscript. It would be good to revise the related paragraph and incorporate this information.

**Claims And Evidence:**

No

**Requested Changes:**

Address [W1]: Please clarify the temporal information and include samples captured over a coarse of the 5-month period (in order) of the same region with labels. Justify the "temporal contrast" and its extension is legitimate given the temporal information of the dataset.

Address [W2]: Clarify the size and statistics (e.g., how many label categories) of the fine-grained segmentation dataset. Justify the results are conclusive given the small dataset or revise this section if the results are not conclusive. Discuss the alternative datasets for fine-grained segmentation in this field. If possible, show results on the alternative datasets.

**Strengths And Weaknesses:**

Strengths
* [S1] As mentioned in the paper, the large-scale benchmark release will unlock opportunities in remote sensing, particularly for agriculture in the community.
* [S2] Experimental results in Table 1 demonstrate the scale of the dataset and its value for self-supervised representation learning. The ResNet-based models are inferior to Swin-T, as the latter is more data-hungry. In addition, the performance of all models has not been saturated (~30% mIoU), which offers future opportunities for self-supervised learning on this benchmark.


Weaknesses
* [W1] The motivations behind Section 4.3-4.4 are very unclear. The reviewer would like to see more details on the temporal information. It is important to include figures illustrating the temporal information and explain how this information can be useful for downstream tasks, especially for an audience who is interested in using AV+ dataset but doesn’t have related domain knowledge. Without knowing this information on the dataset, it is hard to justify the usage of “temporal contrast” and the extension.
* [W2] The writing of section 5.4 is a bit unclear. First, the reviewer cannot tell if “this dataset” is part of AV+ release or not from the first sentence. Second, since the dataset is small, the results on the dataset do not seem to be conclusive. Please elaborate on the “fine-grained semantic segmentation” dataset and compare the dataset with existing ones in the field.

---

> ### Author Response · Authors · 2023-01-10
> **Response to reviewer GJ9C**
>
> We thank the reviewer for acknowledging that our dataset is valuable and proposed benchmarks are detailed. Below please find the responses to some specific comments and questions.
>
> > Please clarify the temporal information and include samples captured over a coarse of the 5-month period (in order) of the same region with labels. Justify the "temporal contrast" and its extension is legitimate given the temporal information of the dataset.
>
> `Answer 1:` We thank the reviewer for requesting further clarification of temporal contrast.  The inclusion of temporal alignment and/or explicit temporal information in remote sensing tasks has been explored in several works, including Ayush et al. [1] and Manas et al. [2].  Because the AV+ dataset includes multiple revisits of the same field, which enables the establishment of positive temporal pairs, we felt it important to include some of these temporal methods in our benchmarks.
>
> We have updated Sections 3.2 and 4.4 of the manuscript with the changes in red and a new figure to demonstrate temporal contrast more clearly.
>
>
> > Clarify the size and statistics (e.g., how many label categories) of the fine-grained segmentation dataset. Justify the results are conclusive given the small dataset, or revise this section if the results are not conclusive. Discuss the alternative datasets for fine-grained segmentation in this field. If possible, show results on the alternative datasets.
>
> `Answer 2:`
>
> As discussed in Section 3.3, we collected 68 flights from the 2020 growth, which are not included in the AV+ dataset. From these flights, 184 tiles with shape 1500$\times$1500 were selected and densely annotated with four classes: soil, weeds, crops, and unmanaged area (e.g., roads, trees, grassed waterways, waterbodies, buildings).
>
> The reviewer rightly states this is a small dataset.  Obtaining densely annotated datasets in agriculture is a major challenge.  Haug et al. [3] created a well-known crop-weed-background(soil) image (ultra high resolution, < 1cm/pixel resolution) dataset with only 60 images.  The 184 images in our experiment took over 500hrs to annotate, which is why exploring SSL methods is so important for this domain.  This experiment was designed to explore transferability to a different task from the same image source (i.e. same camera, same type of fields/crops).  The inclusion of results on this small dataset is designed to demonstrate how the SSL methods transfer to a different, more difficult, real-world task with real-world data limitations, but from the same image source.
>
> To explore the transferability of the AV+ dataset and SSL to other datasets, we elected to focus on the Eurosat Land-Cover task, as that dataset and task are more commonly used in benchmark studies.  Other densely labeled datasets exist in remote sensing data, but not from the same imagery source as AG (i.e. not collected at the same resolution or with the same camera sensors).  For example, Monteiro [4] has a weed-crop-soil dataset for sugar cane collected from drone imagery, while a similar task is at a lower resolution and of a fundamentally different crop. Additionally, image tiles from that dataset are created from an orthomosaic of only a single field, so it is very limited in scope.  With the release of the AV+ dataset and associated benchmarks, we hope other researchers continue to explore its usefulness in a variety of agriculture-focused remote sensing tasks.
>
> We have added more text in red to Section 3.3 to make the description of the task clearer.
>
>
> [1] Ayush, Kumar, Burak Uzkent, Chenlin Meng, Kumar Tanmay, Marshall Burke, David Lobell, and Stefano Ermon. "Geography-aware self-supervised learning." In Proceedings of the IEEE/CVF International Conference on Computer Vision, pp. 10181-10190. 2021.
>
> [2] Manas, Oscar, Alexandre Lacoste, Xavier Giró-i-Nieto, David Vazquez, and Pau Rodriguez. "Seasonal contrast: Unsupervised pre-training from uncurated remote sensing data." In Proceedings of the IEEE/CVF International Conference on Computer Vision, pp. 9414-9423. 2021.
>
> [3] Haug, Sebastian, and Jörn Ostermann. "A crop/weed field image dataset for the evaluation of computer vision based precision agriculture tasks." In European conference on computer vision, pp. 105-116. Springer, Cham, 2015.
>
> [4] Monteiro, A., and A. von Wangenheim. "Orthomosaic Dataset of RGB Aerial Images for Weed Mapping." (2019).

---

### Review · Reviewer_osPe · 2022-12-12

**Summary Of Contributions:**

The paper presents an extended version of the aerial image dataset, Agriculture-Vision, for agricultural pattern classification and segmentation, and builds a benchmark for self-supervised pre-training methods for the agriculture-related downstream tasks. In particular, this work evaluates two backbone networks (ResNet and Swin Transformer) and four different MoCo-based pre-training methods, demontrating their effectiveness on four downstream tasks, including agricultural pattern classification, (fine-grained) semantic segmentation and land-cover classification.

**Audience:**

Yes

**Broader Impact Concerns:**

I have no concerns on the ethical implications.

**Claims And Evidence:**

Yes

**Requested Changes:**

Please address the three concerns in the weaknesses.

**Strengths And Weaknesses:**

Strengths:

+ The proposed dataset augments the original Agriculture-Vision dataset, which includes more than 3600 full-field aerial images collected across 4 years and a small subset with fine-grained segmentation annotations. It is valuable for developing new pre-training methods in remote sensing and fine-grained agricultral pattern analysis.

+ The paper benchmarked a series of MoCo-based pretraining methods, including four variants on two types of network backbones. In particular, it integrates the PPM and/or temporal contrast with MoCo, which provides a detailed study on self-supervised learning strategies from image, pixel and temporal perspective in this application domain.

+ The experimental comparison with the original Agriculture-Vision model demonstrates the superior performance of the methods with pre-training on the segmetnation task.

Weaknesses:

1. Related work:
- Other self-supervised learning techniques have also been explored for semantic segmentation, which was not discussed in this work. While this paper aims to build a generic pre-training model, adding those works could provide a more complete perspective on how the data can be used.  E.g.:
Singh, Suriya, et al. "Self-Supervised Feature Learning for Semantic Segmentation of Overhead Imagery." BMVC. Vol. 1. No. 2. 2018.
Ziegler, Adrian, and Yuki M. Asano. "Self-Supervised Learning of Object Parts for Semantic Segmentation." Proceedings of the IEEE/CVF Conference on Computer Vision and Pattern Recognition. 2022.
- The citation format seems incorrect. For example, in Sec 2.1,  "Specifically pertinent to the current work, Z. Xie et al. (2021) Xie et al. (2021b)  extended the SimCLR framework...". This looks redundant.

2. The proposed SSL study is lacking in several aspects:
- Geo-location information is missing from the representation learning, which seems counter-intuitive. It seems the geography information places an important role in agriculture pattern analysis.
- There is no object-level recognition in the downstream tasks. How well the learned representation would perform on the localization of object instances?
- There is no few-shot learning study on the pre-trained models. What if the labeled data are reduced to a very small amount?

3. Some experimental details are missing. For example, in Sec 5.3.2., the compairson with the supervised method lacks details: Table 2 should include the model sizes for a fair comparison.

---

> ### Author Response · Authors · 2023-01-10
> **Response to reviewer osPe [Part 1/3]**
>
> We thank the reviewer for acknowledging that our dataset is useful for developing new pre-training methods, and superior performance is shown on semantic segmentation tasks. We address the reviewer's concerns as follows:
> > Other self-supervised learning techniques have also been explored for semantic segmentation, which was not discussed in this work. While this paper aims to build a generic pre-training model, adding those works could provide a more complete perspective on how the data can be used. E.g.: Singh, Suriya, et al. "Self-Supervised Feature Learning for Semantic Segmentation of Overhead Imagery." BMVC. Vol. 1. No. 2. 2018. Ziegler, Adrian, and Yuki M. Asano. "Self-Supervised Learning of Object Parts for Semantic Segmentation." Proceedings of the IEEE/CVF Conference on Computer Vision and Pattern Recognition. 2022.
>
> `Answer 1:` We appreciate the reviewer providing these additional references. We will look to incorporate these into the text. We were less familiar with the second paper as it was published less than four months prior to the initial submission of this manuscript.
>
> Importantly, the aim of this paper is _not_ to build a generic pre-training model, but instead to introduce a rich spatiotemporal remote sensing dataset and provide relevant benchmarks.  That is, this is not an algorithms paper; it is a dataset and benchmarks paper.  We have made additions and variations of SSL methods based on MOCO-v2, but this is in the spirit of highlighting the features of the dataset, which may inspire further methodological-focused papers.  SSL is a key topic, with dozens of both general and domain/dataset-specific methods introduced in the past two years alone.  Obtaining benchmarks for every method is not feasible; instead, we have chosen to focus on a single framework (Moco-v2) and captured several variations which can capture the structure and invariances of the high-resolution geo-spatio-temporal dataset we introduce in this work.
>
> > The citation format seems incorrect. For example, in Sec 2.1, "Specifically pertinent to the current work, Z. Xie et al. (2021) Xie et al. (2021b) extended the SimCLR framework...". This looks redundant.
>
> `Answer 2:` We thank the reviewer for catching this- we've made the correction.
>
> > Geo-location information is missing from the representation learning, which seems counter-intuitive. It seems the geography information places an important role in agriculture pattern analysis.
>
> `Answer 3:` We agree with the reviewer that geo-location can be an important piece of information for remote sensing pattern analysis.  Ayush et al. [1] explored the incorporation of locational information for both remote sensing and geo-tagged ImageNet images. Incorporation of location information either through the explicit use of lat/long or through the temporal alignment of the same location often showed improvement over the vanilla MoCo-V2 baseline; the best performing model and significance of the effect varied between datasets and metrics.
>
> The AV+ dataset does not include geo-information; there is subtle location information in the images inasmuch as different images have been identified as belonging to the same parcel at different points in time, but any information which may identify the landowner has been removed.  While we are legally allowed to release the geo-information because the data is collected at a resolution making it part of the public domain, we have elected to error on the side of privacy and not to release this information.  Unlike broad remote sensing datasets, which cover urban areas, forests, agriculture, etc., this data is exclusively focused on managed farmland which is the economic livelihood of the farmers. To be respectful of the privacy of the landowners, we have elected to withhold this information. In discussion with the landowners, they agreed with and were appreciative of this positioning.
>
> Therefore, in this work, we focused on benchmarking the value of the imagery itself following the paradigm of the previous dataset paper [2]. We encourage researchers to explore different methods and utilize other pieces of information like geo-location to benefit machine learning, remote sensing, and the agriculture community. Therefore, thoroughly investigating all the information, like the geo-location of AV+, is out of the scope of this dataset paper.

---

> > ### Author Response · Authors · 2023-01-10
> > **Response to reviewer osPe [Part 2/3]**
> >
> > >There is no object-level recognition in the downstream tasks. How well the learned representation would perform on the localization of object instances?
> >
> > `Answer 4:`  Is the reviewer referring to object detection/instance segmentation in referencing "location of object instances''?  Object (i.e. bounding box) detection for remote sensing agriculture data is an uncommon task for two main reasons:
> >
> > (1) Patterns of instances tend not to be objects, but regions.  For example, give two areas occupied by weeds, how far apart should they be to be considered distinct instances?  Are they even distinct ``instances'' (we would argue "no'' as the designation of a weed cluster is simply an area of weeds with a weed density over a specific threshold)?  Similarly, a class like "drydown'' has no notion of an instance- portions of the field are either driedown or not.  If a grassed-waterway forks into two branches, how many instances are present? And so on.  This is not just a challenge with this particular dataset but is common throughout remote sensing and aerial agriculture.  In agriculture, object detection and instance segmentation are more common in ground-based analysis, such as counting pieces of fruit on a tree, but that is a very different domain from that being explored by this dataset.
> >
> > (2) The number of instances would often be in the hundreds-of-thousands to millions.  For example, given objects which could be counted (e.g. individual plants, if visible), this would number in the millions for a given field.  Therefore methods used to count and/or localize still tend to rely on density-estimation techniques (which are more related to segmentation approaches) instead of discrete bounding box identification [3].
> >
> > As a result of these challenges, semantic segmentation is the more common task in remote sensing for identifying the location of "entities" of interest. Related works such as [1][4][5] do not include detection tasks. In this work, we have performed semantic segmentation for both the AG(supervised) dataset as well as the smaller fine-grained (4-class) semantic segmentation task, which involves locating the precise pixels occupied by a class.
> >
> > Object detection in remote sensing is more common in defense (e.g. planes, tanks, ships) and infrastructure (e.g. buildings, roofs) tasks, which is a very different domain and out of scope for the current work.  DOTA [6] is a detection-focused dataset but excludes agricultural areas and instead includes images of shipping containers, baseball diamonds, basketball courts, planes, vehicles, helicopters, roundabouts, etc.  These classes all have clear definitions of what an instance is and have ~300 instances (max 2000 instances) per image.  DeepGlobe2018 [7] introduces a large dataset that contains both segmentation and detection.  Notably, detection is performed only for building counting, but only semantic segmentation is performed for land-cover identification (which is more closely related to AV+).
> >
> > > There is no few-shot learning study on the pre-trained models. What if the labeled data are reduced to a very small amount?
> >
> > `Answer 5:`  We thank the reviewer for the comments on few-shot learning.  Few-shot learning is an interesting topic related to self-supervised learning, which is worth studying in the context of the AV+ dataset. In fact, it is currently being explored in other parallel investigations.  However, the focus of the paper is the dataset and the demonstration of its value for self-supervised learning through benchmarks. Meanwhile, similar papers usually do not, as a rule, include few-shot experiments[4][8]. Therefore, incorporating the results of few-shot learning is beyond the scope of this paper.
> >
> > > Some experimental details are missing. For example, in Sec 5.3.2., the comparison with the supervised method lacks details
> >
> > `Answer 6:` We have updated Section 5.3.2. in our manuscript to add more experimental details. Changes are highlighted with a different color.
> >
> > >  Table 2 should include the model sizes for a fair comparison.
> >
> > `Answer 7:` We thank the reviewer for the suggestions. We’ve updated Table 3 (i.e., Table 2 in the original version) with the sizes of the models.

---

> ### Author Response · Authors · 2023-01-10
> **Response to reviewer osPe [Part 3/3]**
>
> **References**
>
> [1] Ayush, Kumar, Burak Uzkent, Chenlin Meng, Kumar Tanmay, Marshall Burke, David Lobell, and Stefano Ermon. "Geography-aware self-supervised learning." In Proceedings of the IEEE/CVF International Conference on Computer Vision, pp. 10181-10190. 2021.
>
> [2] Chiu, Mang Tik, Xingqian Xu, Yunchao Wei, Zilong Huang, Alexander G. Schwing, Robert Brunner, Hrant Khachatrian et al. "Agriculture-vision: A large aerial image database for agricultural pattern analysis." In Proceedings of the IEEE/CVF Conference on Computer Vision and Pattern Recognition, pp. 2828-2838. 2020.
>
> [3] Hobbs, Jennifer, Prajwal Prakash, Robert Paull, Harutyun Hovhannisyan, Bernard Markowicz, and Greg Rose. "Large-Scale Counting and Localization of Pineapple Inflorescence Through Deep Density-Estimation." Frontiers in Plant Science 11 (2021): 599705.
>
> [4] Manas, Oscar, Alexandre Lacoste, Xavier Giró-i-Nieto, David Vazquez, and Pau Rodriguez. "Seasonal contrast: Unsupervised pre-training from uncurated remote sensing data." In Proceedings of the IEEE/CVF International Conference on Computer Vision, pp. 9414-9423. 2021.
>
> [5] Vincenzi, Stefano, Angelo Porrello, Pietro Buzzega, Marco Cipriano, Pietro Fronte, Roberto Cuccu, Carla Ippoliti, Annamaria Conte, and Simone Calderara. "The color out of space: learning self-supervised representations for earth observation imagery." In 2020 25th International Conference on Pattern Recognition (ICPR), pp. 3034-3041. IEEE, 2021.
>
> [6] Xia, Gui-Song, Xiang Bai, Jian Ding, Zhen Zhu, Serge Belongie, Jiebo Luo, Mihai Datcu, Marcello Pelillo, and Liangpei Zhang. "DOTA: A large-scale dataset for object detection in aerial images." In Proceedings of the IEEE conference on computer vision and pattern recognition, pp. 3974-3983. 2018.
>
> [7] Demir, Ilke, Krzysztof Koperski, David Lindenbaum, Guan Pang, Jing Huang, Saikat Basu, Forest Hughes, Devis Tuia, and Ramesh Raskar. "Deepglobe 2018: A challenge to parse the earth through satellite images." In Proceedings of the IEEE Conference on Computer Vision and Pattern Recognition Workshops, pp. 172-181. 2018.
>
> [8] Mall, Utkarsh, Bharath Hariharan, and Kavita Bala. "Change Event Dataset for Discovery from Spatio-temporal Remote Sensing Imagery." In Thirty-sixth Conference on Neural Information Processing Systems Datasets and Benchmarks Track.

---

### Review · Reviewer_VzZZ · 2023-01-13

**Summary Of Contributions:**

To push forward agricultural pattern analysis, this paper improves the existing Agriculture-Vision dataset by including raw, full-field imagery. Then, this paper extends this dataset by releasing 3,600 large, high-resolution, full-field, RGB, and near-infrared images, which enable unsupervised pre-training. Next, this paper integrates the Pixel-to-Propagation Module from the SimCLR framework into different contrastive learning approaches for improvement. Finally, this paper benchmarks different self-supervised pre-training methods based on momentum contrastive learning on downstream classification and segmentation tasks. Here, CNN and Swin Transformer are used as backbones.

**Audience:**

Yes

**Broader Impact Concerns:**

The work has no obvious ethical implications.

**Claims And Evidence:**

Yes

**Requested Changes:**

See Weaknesses.

**Strengths And Weaknesses:**

**Strengths**
- The proposed datasets including a pre-training one are novel.
- The experiments are sufficient and convincing.
- The idea is easy to understand.
- The paper's organization is clear.


**Weaknesses**
1. At the human knowledge level, what are the key agricultural patterns, which are concerned with the survival and development of agriculture? Which patterns are reflected in the proposed datasets? To what extent do they help agriculture?

2. The summary of main contributions should be more concise, compact, and precise.

3. The motivations for some contributions are not clear. For example, why does this paper focus on momentum contrastive learning methods rather than other types such as performing pretext tasks and contrasting cluster assignments [a,b,c,d]? Why does this paper incorporate PPM into the MoCo-V2 and SeCo frameworks? How does PPM work? Does such a design achieve the SOTA performance over existing self-supervised learning methods on the proposed datasets? In a word, the introduction section should be self-contained.

[a] Albelwi, S. Survey on Self-Supervised Learning: Auxiliary Pretext Tasks and Contrastive Learning Methods in Imaging. Entropy 2022, 24, 551. https://doi.org/10.3390/e24040551.

[b] Qing Chang, Junran Peng, Lingxi Xie, Jiajun Sun, Haoran Yin, Qi Tian, Zhaoxiang Zhang; DATA: Domain-Aware and Task-Aware Self-Supervised Learning; Proceedings of the IEEE/CVF Conference on Computer Vision and Pattern Recognition (CVPR), 2022, pp. 9841-9850.

[c] Adrian Ziegler, Yuki M. Asano; Self-Supervised Learning of Object Parts for Semantic Segmentation; Proceedings of the IEEE/CVF Conference on Computer Vision and Pattern Recognition (CVPR), 2022, pp. 14502-14511.

[d] A. Khan, S. AlBarri and M. A. Manzoor, "Contrastive Self-Supervised Learning: A Survey on Different Architectures," 2022 2nd International Conference on Artificial Intelligence (ICAI), Islamabad, Pakistan, 2022, pp. 1-6, doi: 10.1109/ICAI55435.2022.9773725.

4. Is it enough for downstream transferring to pre-train the model on only 3,600 images? As we know, the ImageNet dataset has 1.28M natural images. Thus, a problem is raised naturally: how does the pre-training data quantity affect the downstream classification or segmentation performance? Why don't the authors collect more full-field images for pre-training?

5. At the pre-training stage, the pixel-level loss is computed between the smoothed q_i^s from PPM and k_j. Without any annotations, how can it be sure that pixels i and j are a positive pair?

6. The data augmentation operations used in MoCo-V2 are tailored for natural images. Would they be fit for remote sensing imagery? How does remote sensing imagery differ from natural images in terms of appearance and structure? I wonder whether there are data augmentations designed for the characteristics of remote sensing imagery.

7. In Table 1, with ResNet as the backbone and fine-tuning as the downstream training strategy, MoCo-PixPro performs worse than MoCo-V2, why does PPM yield a negative effect?

8. The line number should be shown for ease of reference!

9. Grammar and symbol errors. The authors should carefully proofread the paper and revise the errors. For example, at line 5 of paragraph 1 in Sec. 1, 'exists' -> 'exist'; at line 8 of paragraph 3 in Sec. 1, 'agriculture dataset' -> 'agriculture datasets'; at line 9 of paragraph 3 in Sec. 1, 'which is' -> 'which are'; at last line of paragraph 2 in Sec 3.3, 'resemble' -> 'resembles'; at line 7 of paragraph 1 in Sec. 4.1, 'q and k' -> 'k+ and k-'; at line 5 under Eq. (3), 'features maps' -> 'feature maps'; at line 1 of paragraph 1 in Sec 5.1, 'follow' -> 'following'.

---

> ### Author Response · Authors · 2023-01-23
> **Response to reviewer VzZZ [1/3]**
>
> We thank the reviewer for acknowledging that our data with benchmarks for pre-training is novel, and experiments are sufficient and convincing. We hereby answer the reviewer's questions:
>
> > At the human knowledge level, what are the key agricultural patterns which are concerned with the survival and development of agriculture? Which patterns are reflected in the proposed datasets? To what extent do they help agriculture?
>
> `Answer 1:` The patterns of this paper were selected by the authors of Chiu et al. 2020 [1] under the advisement of agronomists. All the patterns either reflect the status of crops or play the role of early warnings with an emphasis on sustainable agriculture, thus are all critical for farmers and agricultural researchers. For example, the selected categories, such as **nutrient deficiency** and **drydown**, capture a key health indicator and growth stage, respectively. For further details of each pattern, please find them in the supplementary materials.
>
> > The summary of main contributions should be more concise, compact, and precise.
>
> `Answer 2:` We thank the reviewer's suggestions. We've modified the summary accordingly.
>
> > The motivations for some contributions are not clear. For example, why does this paper focus on momentum contrastive learning methods rather than other types such as performing pretext tasks and contrasting cluster assignments [a,b,c,d]? Why does this paper incorporate PPM into the MoCo-V2 and SeCo frameworks? How does PPM work? Does such a design achieve the SOTA performance over existing self-supervised learning methods on the proposed datasets? In a word, the introduction section should be self-contained.
>
> `Answer 3:` We appreciate the reviewer providing these additional references; we will look to incorporate these into the discussion of this paper.
>
> Notably, this paper aims to publish the dataset and benchmark the dataset using the most widely used contrastive learning method. Among different contrastive learning models, we chose MoCo given its robustness and achievable batch size to broader researchers. Through pre-training, we adapted the pre-trained backbones/encoders to different sub-tasks and achieved promising performances. These results show the effectiveness of pre-training with contrastive learning. However, the investigation of the unsupervised learning/contrastive learning model is not the focus of this work. What is more important is to demonstrate the great potential of AV+ that benefits various learning tasks of multiple domains, even with some straightforward pre-trainings.
>
> While there are many unsupervised and semi-supervised methods, we additionally explore variants of pre-training methods based on the laid nature of AV+. Unlike traditional datasets like ImageNet, AV+ has more abundant semantic meaning at the pixel level. In other words, a few pixels in AV+ could represent one pattern that is critical for representation learning. Therefore, we introduce PPM to emphasize pixel-level contrast. The technical details of PPM can be found in Section 4.2. Another characteristic of remote sensing is data revisiting, i.e., capturing images from the same locations at different times. To enable the temporal difference(temporal contrast) from data revisiting to serve as an additional variation for pre-training, we adapt the multi-head structure from SeCo.
>
> It is fair to claim that the proposed methods achieve the SOTA based on the publication records, but we'd rather view our models as benchmarks and a pilot study of unsupervised learning on AV+ following the paradigm of the previous dataset paper [1]. We encourage researchers to explore other methods to benefit machine learning, remote sensing, and the agriculture community.
>
> > At the pre-training stage, the pixel-level loss is computed between the smoothed $q_i^s$ from PPM and $k_j$. Without any annotations, how can it be sure that pixels i and j are a positive pair?
>
> `Answer 4:` To decide positive pairs for contrast, each feature map is first warped to the original image space. Then the distances between the pixel $i$ and pixel $j$ from each of the two feature maps are computed and normalized. Given a hyper-parameter $\tau$ (set as 0.7 by default), $i$ and $j$ are recognized as one positive pair if their distance is less than $\tau$. We've also updated the explanation to Section 4.2.2 in red.

---

> > ### Author Response · Authors · 2023-01-23
> > **Response to reviewer VzZZ [2/3]**
> >
> > > The data augmentation operations used in MoCo-V2 are tailored for natural images. Would they be fit for remote sensing imagery? How does remote sensing imagery differ from natural images in terms of appearance and structure? I wonder whether there are data augmentations designed for the characteristics of remote sensing imagery
> >
> > `Answer 5:` We conducted experiments with combinations of different data augmentation methods. Based on our results, the data augmentation pipelines used in the original MoCo are still optimal. This observation is consistent with previous research on MoCo in remote sensing [2]. We thank the reviewer for coming up with this question and will update this observation in Section 5.1.
> >
> > > In Table 1, with ResNet as the backbone and fine-tuning as the downstream training strategy, MoCo-PixPro performs worse than MoCo-V2, why does PPM yield a negative effect?
> >
> > `Answer 6:` While MoCo with PPM benefits the learning of embeddings for the downstream segmentation task, it is not necessarily true when fine-tuning the models. One important observation is that the PPM benefits more as we scale up the models from ResNet-18 to ResNet-50 and then Swin-T. As the training epochs are all the same for all the pre-training, smaller backbones like ResNet-18 are more likely to get overfitted. When trained with ResNet-50, the performance drop of MoCo-PixPro is very small compared with MoCo. As we move to Swin-T, MoCo-PixPro eventually shows the best performance over other methods. We've added this discussion in 5.3.1.
> >
> > > The line number should be shown for ease of reference
> >
> > `Answer 7:` We thank the reviewer's suggestions. We've added the line number for better reference.
> >
> > > Grammar and symbol errors. The authors should carefully proofread the paper and revise the errors. For example, at line 5 of paragraph 1 in Sec. 1, 'exists' -> 'exist'; at line 8 of paragraph 3 in Sec. 1, 'agriculture dataset' -> 'agriculture datasets'; at line 9 of paragraph 3 in Sec. 1, 'which is' -> 'which are'; at last line of paragraph 2 in Sec 3.3, 'resemble' -> 'resembles'; at line 7 of paragraph 1 in Sec. 4.1, 'q and k' -> 'k+ and k-'; at line 5 under Eq. (3), 'features maps' -> 'feature maps'; at line 1 of paragraph 1 in Sec 5.1, 'follow' -> 'following'.
> >
> > `Answer 8:`We thank the reviewer for pointing out these typos. We've updated the paper to fix these.
> >
> >
> > **References**
> >
> > [1] Chiu, Mang Tik, Xingqian Xu, Yunchao Wei, Zilong Huang, Alexander G. Schwing, Robert Brunner, Hrant Khachatrian et al. "Agriculture-vision: A large aerial image database for agricultural pattern analysis." In Proceedings of the IEEE/CVF Conference on Computer Vision and Pattern Recognition, pp. 2828-2838. 2020.
> >
> > [2] Manas, Oscar, Alexandre Lacoste, Xavier Giró-i-Nieto, David Vazquez, and Pau Rodriguez. "Seasonal contrast: Unsupervised pre-training from uncurated remote sensing data." In Proceedings of the IEEE/CVF International Conference on Computer Vision, pp. 9414-9423. 2021.

---

> ### Comment · Reviewer_VzZZ · 2023-02-06
> **An issue unsolved!**
>
> The 4-th issue in Weaknesses has not been addressed.

---

> > ### Author Response · Authors · 2023-02-06
> > **Response to reviewer VzZZ [3/3]**
> >
> > We thank the reviewer's reminder. Below please find the response to the 4th Weakness.
> >
> >
> > > Is it enough for downstream transferring to pre-train the model on only 3,600 images? As we know, the ImageNet dataset has 1.28M natural images. Thus, a problem is raised naturally: how does the pre-training data quantity affect the downstream classification or segmentation performance? Why don't the authors collect more full-field images for pre-training?
> >
> > `Answer:` As demonstrated in Figure 1, a typical raw image has a shape of $15,000 \times 15,000$. If we crop all the raw images into the average resolution of ImageNet, i.e., $469 \times387$, we eventually obtain over 4M images. We add a further description of the raw images in Section 3.2 to address the reviewer's concern. To evaluate the influence of data quantity on downstream tasks, we report the ablation study in section 5.6.

---

### Review · Reviewer_TPh3 · 2023-01-17

**Summary Of Contributions:**

The paper extends the original Agriculture-Vision (AV) dataset with full-field labeled images, full-field unlabeled images for pre-training, and finer-grained labeled images for segmentation. (contribution on datasets) The authors then evaluate different pre-training strategies (together with different model architectures) on the new dataset AV+. (contribution on benchmarking).

**Audience:**

Yes

**Claims And Evidence:**

No

**Requested Changes:**

Please see the above weaknesses.

Additionally, I strongly encourage the authors to show some images that contain all nine patterns and clearly illustrate the difference between the segmentation data in AV+ and the fine-grained segmentation data. I also strongly encourage the authors to create a table that clearly summarizes the experimental setups.


**Strengths And Weaknesses:**

#Strength#

S1. The paper considers both classification and segmentation tasks and considers an even fine-grained segmentation task.

S2. The paper considers multiple neural network architectures and multiple pre-training strategies.

#Weakness#

**W1. Writing needs significant improvement.** There are many typos and grammatical errors, such as “in symmetric form”, “are follow”, “performs best”, “AG+ or AV+” just to point out a few. I would strongly recommend that the authors run a Grammarly check. The citation format also can be improved --- please use \citep, \citet, \cite appropriately. Also, there are some logically inconsistent sentences, such as: *"Many approaches originally developed for natural images work well on remote sensing imagery with only minimal modification, although this is not guaranteed due to the large domain gap. Additionally, they may fail to exploit the unique structure of earth observation data such as …"* The writing issue makes it a bit hard to read the paper.

**W2. The background information, approach, and experimental setup are not well organized and presented.**

W2.1. Figure 2 is not self-contented. There are no clear meanings for the use of colors. Also, the same block shapes sometimes refer to parameters or to losses, making the figure a bit confusing.

W2.2. The loss in sect. 4.2 is a bit hard to grasp. First, $y_i$ is defined but not used. Second, how is the pair $(i, j)$ defined? Are $i$ and $j$ the same pixels but at different pixel locations due to augmentation?

W2.3. It is a bit hard to understand the usefulness/goal of temporal contrast. Do the authors expect features across different times to be invariant? However, at different times, the patterns/labels to be recognized may change.

W2.4. Fig 3 and Figures 4 and 5 are not in the same style.

W2.5. Perhaps the severest issue of the paper is the lack of a clear description of the experimental setups. **First**, given that AV+ provides the full-field images, are the pre-trained and downstream models trained upon tiles or the full-field images? What are their sizes (height x width)? How many full-field labeled images are there? (94986, or more)? How are these images/tiles separated into train/test/validation? Is there a guarantee that nearby images (or images at the same geo-location) won’t be included in both train and test sets? Is the unlabeled set (2019-2020) collected at the same locations as the labeled set (2017-2019)? Logically, it might be a bit strange to use earlier images for testing/downstream tasks, and later images for pre-training. **Second**, there is no detailed description of the training and test protocols. **Third**, the difference between the "normal" segmentation data in AV+ and those in the "fine-grained" set is not clear. Does the "higher resolution" refer to the input image or just the output labeling? Also, how are the ground-truth labels obtained? If it is by humans, is there a carefully-designed way to guarantee quality? **Finally**, suppose the numbers of categories in normal fine-grained and fine-grained segmentation are the same (9 categories), it is unclear to me why only the latter encounters label imbalance and needs the focal loss.

**W3. Lack of contributions and insights.**

W3.1. To me, the extension of MoCo-V2 with Seasonal Contrast and PPM seems to be quite straightforward (i.e., there seem to be no difficulties in combining them). Thus, this contribution is quite marginal.

W3.2. I would hope to see an analysis of how much the NIR channel contributes to the final performance, but there is no such analysis.

W3.3. In sect. 5.4, there is no further discussion and analysis on why the ImageNet pre-trained model leads to the best performance. Also, the argument on larger models *"The ResNet-50 and Swin-T models performed relatively worse compared to the ResNet-18 models which is unsurprising given the extremely small size of this dataset."* seems not convincing --- the authors already use a large dataset for pre-training. If the issue is in the decoder, the authors may consider a supervised pre-training step for the decoder using the segmentation data in AV+.

W3.4. Results in Table 2 seem to be not comparable – different training strategies and model architectures are used. Also, did the original AV paper use the NIR channel?

W3.5. There is no further discussion and comparison on how the full-field images help the classification or segmentation. As the release of the full-field images seems to be a contribution, it would be great to see such a comparison.

W3.6. Except for showing that PPM improves the pre-training quality, there seem to be limited insights from this paper. I would hope to see some more discussions on why the temporal contrast does not consistently improve the performance. I would also want to see more discussions on how this paper improves scientific understanding, e.g., in agriculture.

---

> ### Author Response · Authors · 2023-01-27
> **Response to reviewer TPh3 [1/5]**
>
> We thank the reviewer for acknowledging that our paper considers multiple sub-tasks, model architectures and pre-training strategies. Below please find the responses to your questions and we sincerely hope they could address your concerns:
>
> >W1. Writing needs significant improvement. There are many typos and grammatical errors, such as “in symmetric form”, “are follow”, “performs best”, “AG+ or AV+” just to point out a few. I would strongly recommend that the authors run a Grammarly check. The citation format also can be improved --- please use cite, citp, citet appropriately. Also, there are some logically inconsistent sentences, such as: "Many approaches originally developed for natural images work well on remote sensing imagery with only minimal modification, although this is not guaranteed due to the large domain gap. Additionally, they may fail to exploit the unique structure of earth observation data such as …" The writing issue makes it a bit hard to read the paper.
>
> `Answer 1:` We thank the reviewer for pointing out these issues. We've corrected them accordingly. We will also clarify the logic in Section 1, written in red.
>
> > W2.1. Figure 2 is not self-contented. There are no clear meanings for the use of colors. Also, the same block shapes sometimes refer to parameters or to losses, making the figure a bit confusing.
>
> `Answer 2.1:` The reason for assigning different colors to the blocks in Figure 2 is to clarify the different purposes of the corresponding modules in the overall framework. To be more specific, modules in navy blue serve as encoders for feature extraction. Modules in brown are designed for pixel contrast, which includes projectors, pixel propagation modules and the loss is used. Pink modules represent instance-level contrast with embeddings space invariant to all kinds of augmentations. Similarly, modules in green and sky blue mean instance-level contrast but extract features invariant to artificial augmentation and temporal augmentation, respectively. We have updated these details in Section 4.4 in red.
>
> > W2.2. The loss in sect. 4.2 is a bit hard to grasp. First,  is defined but not used. Second, how is the pair defined? Are and  the same pixels but at different pixel locations due to augmentation?
>
> `Answer 2.2:` We have updated the notation from $y_{i}$ to $q_{i}^{s}$,  in equation(2) and equation (3) for a clearer explanation of loss computation. Generally, it is correct to think of $i$ and $j$ as the same pixel in the original image. For more details, the reviewer can refer to the added explanations of the positive pair($i$, $j$) in Section 4.2.2 in red.
>
>
> > W2.3. It is a bit hard to understand the usefulness/goal of temporal contrast. Do the authors expect features across different times to be invariant? However, at different times, the patterns/labels to be recognized may change.
>
> `Answer 2.3:` We thank the reviewer for requesting further clarification of temporal contrast. The inclusion of temporal alignment and/or explicit temporal information in remote sensing tasks has been explored in several works, including Ayush et al. [1] and Manas et al. [2]. The notion of temporal contrast was introduced by Manas et al. [2] in 2021 and is a leading approach for self-supervised learning for remote sensing and earth observation data; accordingly, we included it in our set of benchmarks.  This approach takes advantage of the time and positional invariance in these datasets to learn transferable representations for remote sensing applications.
>
> Notably, the temporal gap between positive pairs varies from one to six months. Most patterns, such as Water, Nutrient Deficiency, Waterways and Weeds, are kept consistent/invariant in the time frame of data collection, as illustrated in Figure 3.   Because the AV+ dataset includes multiple revisits of the same field, which enables the establishment of positive temporal pairs, we felt it important to include some of these temporal methods in our benchmarks.
>
> > W2.4. Fig 3 and Figures 4 and 5 are not in the same style.
>
> `Answer 2.4:` Could the reviewer kindly elaborate on what style differences are observed?  The figures were created with the same style format, so perhaps something has tripped up in the render.  We will address this if the reviewer could provide additional detail.

---

> > ### Author Response · Authors · 2023-01-27
> > **Response to reviewer TPh3 [2/5]**
> >
> > > W2.5.1 First, given that AV+ provides the full-field images, are the pre-trained and downstream models trained upon tiles or the full-field images? What are their sizes (height x width)? How many full-field labeled images are there? (94986, or more)? How are these images/tiles separated into train/test/validation? Is there a guarantee that nearby images (or images at the same geo-location) won’t be included in both train and test sets? Is the unlabeled set (2019-2020) collected at the same locations as the labeled set (2017-2019)? Logically, it might be a bit strange to use earlier images for testing/downstream tasks, and later images for pre-training.
> >
> > `Answer 2.5.1:` We thank the reviewer for asking these detailed questions. Many of the details surrounding the supervised portion of the data are given in Chiu et al. [3], and we would refer the reviewer to those details.  We have also clarified the text to enhance these points.  For pre-training, we have added the suggested experimental details in Section 5.1 in red. For labeled images, there are a total of 94,986 tiles with a size of $512 \times 512$, as described in Section 3.1. With a 6/2/2 train/val/test ratio, the Agriculture-Vision dataset contains 56,944/18,334/19,708 train/val/test images. We have re-stated this information in red in Section 5.2 and Section 5.3 for the classification and segmentation tasks, respectively. Based on the description of [3], no nearby images from the Agriculture-Vision dataset are collected in both train and test sets, as all the images are collected from different farmlands. There is also no overlapping between the unlabeled parts of AV+ and the original Agriculture-Vision dataset.
> >
> > We also understand that Chiu et al. [3] omitted a number of details about the dataset that researchers could find useful.  Therefore we have created (included in the original submission) a description of many of these patterns in the supplemental material for those who are interested.
> >
> > > W2.5.2 Second, there is no detailed description of the training and test protocols.
> >
> > `Answer 2.5.2:` Could the reviewer kindly elaborate on what information they believe is missing?  In Section 5.1, "Pre-training Settings," we describe the various parameters used for pre-training, including the batch size, number of epochs, weight decay, momentum, learning rate, etc. Parameters for linear probing, non-linear probing, and fine-tuning are given in their respective sections 5.2.1 - 5.2.2, respectively.  Similar values are given in 5.3, 5.4, and 5.5.
> >
> > > W2.5.3 Third, the difference between the "normal" segmentation data in AV+ and those in the "fine-grained" set is not clear. Does the "higher resolution" refer to the input image or just the output labeling? Also, how are the ground-truth labels obtained? If it is by humans, is there a carefully-designed way to guarantee quality?
> >
> > `Answer 2.5.3:` The "normal'' segmentation task the reviewer is referring to is based on the original Agriculture-Vision dataset [3].  As described in that work, that dataset contains annotations of patterns like storm damage, planter skip, and water. However, while a field may have multiple classes present, the vast majority of the 94,986 images have only a single class present.  Furthermore, a class like ``weed cluster'' captures a region of the field with a weed density over a certain threshold.  However, this area may contain weeds, crop, and soil.  We realize this is confusing, which is why we have provided an extended description of this in the supplemental, but this is how those authors defined the dataset.  The annotation and QA methodology are described in detail in [3].
> >
> > In contrast, the ``fine-grained segmentation task'' is a fundamentally new set of data and patterns that were not included in [3].  The resolution is still the same (10cm/pixel). Here, every single pixel is labeled as weed, crop, soil or unmanaged.  Additionally, this dataset is very small, with only 184 tiles. This is a seemingly trivial task; however, since each image takes 2-3hrs to annotate and QA in this manner because of the intricacy of the crop structure and the smallness of the weeds and crop (perhaps only 1-2 pixels in size), it is extraordinarily difficult to collect. Therefore the hope is that self-supervised learning would be useful in enabling good performance on this real-world task.  These images went through two rounds of human annotation and QA by domain experts.

---

> > > ### Author Response · Authors · 2023-01-27
> > > **Response to reviewer TPh3 [3/5]**
> > >
> > > > W2.5.4 Finally, suppose the numbers of categories in normal fine-grained and fine-grained segmentation are the same (9 categories), it is unclear to me why only the latter encounters label imbalance and needs the focal loss.
> > >
> > > `Answer 2.5.4:` We thank the reviewer for considering such an assumption in our experiments. However, the "fine-grained segmentation task'' must have four classes as it is formulated. Every pixel can belong to one and only one of the four classes since this is dictated by agronomy. In the Agriculture-Vision dataset, some classes could be mutually exclusive, but others are not required to be so. For example, there could be a weed cluster growing in standing water due to recent flooding. This is rare and may not be observed, but nothing prevents it from being so. There could be nutrient deficiency within the endrows of a field. Again, we believe this was not well explained in [3], which is why we have taken the time to elaborate on these relationships in the supplemental material.
> > >
> > > > W3. Lack of contributions and insights.
> > >
> > > `Answer 3:` We thank the reviewer for their time in providing feedback. We would like to reiterate here what we believe are the key contributions of this work:
> > >
> > > [$\bullet$] We release a full-field version of the Agriculture-Vision dataset to further encourage broad agricultural research in pattern analysis
> > >
> > >  [$\bullet$] We release over 3 terabytes of unlabeled, full-field images from more than 3600 full-field images to enable unsupervised pre-training.
> > >
> > > [$\bullet$] We benchmark self-supervised pre-training methods based on momentum contrastive learning and evaluate their performance on downstream classification and semantic segmentation tasks with variable amounts of annotated data.
> > >
> > > [$\bullet$] We perform benchmarks using both CNN and Swin Transformer backbones.
> > >
> > > [$\bullet$] We incorporate the Pixel-to-Propagation Module  (PPM) [9], originally built on SimCLR [7], into the MoCo-V2 [8] and evaluate its performance.
> > >
> > > [$\bullet$] We adapt the approach of Seasonal Contrast (SeCo) [2] for this dataset, which contains imagery only during the growing season to address the spatiotemporal nature of the raw data specifically.
> > >
> > >
> > > > W3.1. To me, the extension of MoCo-V2 with Seasonal Contrast and PPM seems to be quite straightforward (i.e., there seem to be no difficulties in combining them). Thus, this contribution is quite marginal.
> > >
> > > `Answer 3.1:` The aim of this paper is not to build a generic pre-training model, but instead to introduce a rich spatiotemporal remote sensing dataset and provide relevant benchmarks. That is, this is not an algorithms paper; it is a dataset and benchmarks paper. We have made additions and variations of SSL methods based on MOCO-v2, but this is in the spirit of highlighting the features of the dataset, which may inspire further methodological-focused papers.
> > >
> > > While the extension of MoCo-V2 SeCo and PPM is intuitively straightforward, the effectiveness of these models in remote sensing is unexplored and requires massive empirical experiments. Additionally, incorporating Swin-T into the framework of MoCo is non-trivial. Previous work [4] migrating MoCo's backbones from ResNet to Vision Transformer [5] is sufficient for an independent publication. We follow the same paradigm to explore the Swin Transformer's performance with MoCo since this is not yet uncovered in previous research to the best of our knowledge.
> > >
> > > > W3.2. I would hope to see an analysis of how much the NIR channel contributes to the final performance, but there is no such analysis.
> > >
> > > `Answer 3.2:` We thank the reviewer for the suggestion. We've updated Table 2 to illustrate the model's performance. A corresponding discussion of these results is added in Section 5.3.2.
> > >
> > > > W3.3. In sect. 5.4, there is no further discussion and analysis on why the ImageNet pre-trained model leads to the best performance. Also, the argument on larger models "The ResNet-50 and Swin-T models performed relatively worse compared to the ResNet-18 models which is unsurprising given the extremely small size of this dataset." seems not convincing --- the authors already use a large dataset for pre-training. If the issue is in the decoder, the authors may consider a supervised pre-training step for the decoder using the segmentation data in AV+.
> > >
> > > `Answer 3.3:` ImageNet weights are perhaps shockingly effective in a domain as different as remote sensing data.  Other works, including Manas et al. [2], also saw instances where Imagenet weights outperformed their SSL weights on different experiments (see [2] Table 4).
> > >
> > > We agree that also pre-training the decoder could lead to improved performance.  However, in SSL works such as PixPro [3], which include performance on dense tasks (which is much less common), only the encoder is pre-trained for dense tasks such as semantic or instance segmentation.  For a benchmark paper, we have elected to follow this common practice.

---

> > > > ### Author Response · Authors · 2023-01-27
> > > > **Response to reviewer TPh3 [4/5]**
> > > >
> > > > > W3.4. Results in Table 2 seem to be not comparable – different training strategies and model architectures are used. Also, did the original AV paper use the NIR channel?
> > > >
> > > > `Answer 3.4:` For this analysis, we have used the same settings to have a fair comparison with the previous SOTA. We have added the training details for this comparison in Section 5.3.2. Importantly, we use the smaller U-Net in this segmentation task compared to previous FPN and DeeplabV3 models. This would seem like a disadvantage. However, the U-Net is competitive without any extra hyper-parameter search. We feel such a setting is critical since it shows the value of the AV+: Representations pre-trained from AV+ are powerful and can be easily generalized to multiple sub-tasks without much effort.
> > > >
> > > > We have updated Table 2 to provide further information about the NIR channel for the original and our paper.
> > > >
> > > > Again, we'd emphasize the goal of this work is not to discover the optimal model architecture for the supervised Agriculture-Vision dataset.  This is a dataset and benchmark paper, with the contribution being the release of a large spatiotemporal remote sensing dataset and the establishment of benchmarks around using this dataset for pertaining.
> > > >
> > > > > W3.5. There is no further discussion and comparison on how the full-field images help the classification or segmentation. As the release of the full-field images seems to be a contribution, it would be great to see such a comparison.
> > > >
> > > > `Answer 3.5:` We thank the reviewer for recognizing the importance of this contribution.  The original agriculture vision dataset is published with all images cropped into a size of 512 $\times$ 512. While this pre-processing ensures consistency with datasets of natural images, it causes some information loss of geo-relationships among cropped images. Researchers devoted significant efforts to exploring such information by stitching cropped images back to the raw format [6]. While such a method is effective for learning tasks, it will take many efforts and sometimes fail because of missing tiles. Therefore, we publish the raw full-field imagery to fix this issue and encourage researchers to continue to explore such a geo-information at different scales.  The value in the added flexibility of being able to introduce more variability through different cropping strategies has been seen at the CVPR Agriculture-Vision Workshop Prize Challenge with the winners over past several years employing this strategy; these participants stitched together the original imagery, then cut it up in different ways to introduce more data.  We hope releasing the data in this fashion will eliminate this unnecessary preprocessing step for researchers in the future.
> > > >
> > > > Additionally, the use of newer architectures that enable the training and inference of the full field (which can be more than 1GB in size) on a single GPU is the focus of ongoing work.
> > > >
> > > >
> > > > > W3.6. Except for showing that PPM improves the pre-training quality, there seem to be limited insights from this paper. I would hope to see some more discussions on why the temporal contrast does not consistently improve the performance. I would also want to see more discussions on how this paper improves scientific understanding, e.g., in agriculture.
> > > >
> > > > `Answer 3.6:` The important difference between our TemCo and SeCo is the time difference. While the time gap of images AV+ varies from 1 week to 6 months, SeCo uses a dataset with a three-month time gap. In other words, AV+ post more challenging pre-training task as it requires the backbones to capture temporal features that are variant/invariant at multiple time scales. Therefore, the performance of TemCo/SeCo is expected. We chose to report the results to motivate researchers to explore other methods/models by taking advantage of this temporal information laid in AV+.
> > > >
> > > > The importance of agriculture is detailed in Chiu et al. [3].  In the present work, the contribution is that the use of this raw data combined with appropriate SSL techniques enables obtaining those insights with less supervised data (or provides better results for the same amount of labeled data).  However, we thank the reviewer's suggestions. We will add additional text to the introduction/conclusion to emphasize the impact automated pattern identification from remote sensing has on agriculture.

---

> > > > > ### Author Response · Authors · 2023-01-27
> > > > > **Response to reviewer TPh3 [5/5]**
> > > > >
> > > > > **References**
> > > > >
> > > > > [1] Ayush, Kumar, Burak Uzkent, Chenlin Meng, Kumar Tanmay, Marshall Burke, David Lobell, and Stefano Ermon. "Geography-aware self-supervised learning." In Proceedings of the IEEE/CVF International Conference on Computer Vision, pp. 10181-10190. 2021.
> > > > >
> > > > > [2] Manas, Oscar, Alexandre Lacoste, Xavier Giró-i-Nieto, David Vazquez, and Pau Rodriguez. "Seasonal contrast: Unsupervised pre-training from uncurated remote sensing data." In Proceedings of the IEEE/CVF International Conference on Computer Vision, pp. 9414-9423. 2021.
> > > > >
> > > > > [3] Chiu, Mang Tik, Xingqian Xu, Yunchao Wei, Zilong Huang, Alexander G. Schwing, Robert Brunner, Hrant Khachatrian et al. "Agriculture-vision: A large aerial image database for agricultural pattern analysis." In Proceedings of the IEEE/CVF Conference on Computer Vision and Pattern Recognition, pp. 2828-2838. 2020.
> > > > >
> > > > > [4] Chen, Xinlei, Saining Xie, and Kaiming He. "An empirical study of training self-supervised vision transformers." In Proceedings of the IEEE/CVF International Conference on Computer Vision, pp. 9640-9649. 2021.
> > > > >
> > > > > [5] Dosovitskiy, Alexey, Lucas Beyer, Alexander Kolesnikov, Dirk Weissenborn, Xiaohua Zhai, Thomas Unterthiner, Mostafa Dehghani et al. "An image is worth 16x16 words: Transformers for image recognition at scale." arXiv preprint arXiv:2010.11929 (2020).
> > > > >
> > > > > [6] Chiu, Mang Tik, Xingqian Xu, Kai Wang, Jennifer Hobbs, Naira Hovakimyan, Thomas S. Huang, and Honghui Shi. "The 1st agriculture-vision challenge: Methods and results." In Proceedings of the IEEE/CVF Conference on Computer Vision and Pattern Recognition Workshops, pp. 48-49. 2020.
> > > > >
> > > > > [7] Chen, Ting, Simon Kornblith, Kevin Swersky, Mohammad Norouzi, and Geoffrey E. Hinton. "Big self-supervised models are strong semi-supervised learners." Advances in neural information processing systems 33 (2020): 22243-22255.
> > > > >
> > > > > [8] Chen, Xinlei, Haoqi Fan, Ross Girshick, and Kaiming He. "Improved baselines with momentum contrastive learning." arXiv preprint arXiv:2003.04297 (2020).
> > > > >
> > > > > [9] Xie, Zhenda, Yutong Lin, Zheng Zhang, Yue Cao, Stephen Lin, and Han Hu. "Propagate yourself: Exploring pixel-level consistency for unsupervised visual representation learning." In Proceedings of the IEEE/CVF Conference on Computer Vision and Pattern Recognition, pp. 16684-16693. 2021.

---

> > ### Comment · Reviewer_TPh3 · 2023-02-13
> > **Further comments to [1/5]**
> >
> > **To Answer 2.1:** Please add the text to your Figure 2 caption.
> >
> > **To Answer 2.4:** My bad. It should have been Figure 4 vs. Figure 5 and Figure 6. The former is in a bar plot, while the latter two are in curve plots. I'd suggest that the authors use just one style.
> >
> > **To Answer 2.5.1:** I really don't buy the reason why the authors didn't include experimental/data collection details. As a reader, I would expect to read a paper and learn how to reproduce it, rather than needing to refer to another paper, not to mention that new datasets are considered contributions to this paper. The authors should have a table to clearly summarize the data statistics. Please also note that for the places I have questions about, the authors did not have a clear pointer that some information can be found in the supplementary, even in their refined version.
> >
> > **Answer 2.5.2:** The testing protocol question was mainly about how to prepare test data (tiles or full images) and how many they are. It is about if the reader is able to reproduce the experiments.

---

> > > ### Author Response · Authors · 2023-02-15
> > > **Reply to "Further comments to [1/5]"**
> > >
> > > > Please add the text to your Figure 2 caption.
> > >
> > > `Answer:`
> > > We thank the reviewer's suggestion. We've moved the updated details from Section 4.4 to the caption of Figure 2 in red.
> > >
> > > > It should have been Figure 4 vs. Figure 5 and Figure 6. The former is in a bar plot, while the latter two are in curve plots. I'd suggest that the authors use just one style.
> > >
> > > `Answer:`
> > > We thank the reviewer's clarification. We've converted the bar plot to curve plots for consistency, providing further data points.
> > >
> > > > I really don't buy the reason why the authors didn't include experimental/data collection details. As a reader, I would expect to read a paper and learn how to reproduce it, rather than needing to refer to another paper, not to mention that new datasets are considered contributions to this paper. The authors should have a table to clearly summarize the data statistics. Please also note that for the places I have questions about, the authors did not have a clear pointer that some information can be found in the supplementary, even in their refined version.
> > >
> > > `Answer:`
> > > We understand the reviewer's concern as a reader. Therefore, we continue to add one more table to compare the statistics of the original AV and the proposed AV+ in Section 3.2. As Chiu et al. [3] omitted a number of details about the dataset that researchers could find useful, we have created a description of many of these patterns in the supplemental material for those who are interested in supplementary. To be specific, please find the pattern-related information in Section 1.3 (Class labels: Label semantics). If the reviewer thinks other information is missing/required, we can provide further explanation or clarification accordingly.
> > >
> > > > The testing protocol question was mainly about how to prepare test data (tiles or full images) and how many they are. It is about if the reader is able to reproduce the experiments.
> > >
> > > `Answer:`
> > > We thank the reviewer's clarification again. We add additional sections under Section 5.2 and Section 5.3 noted as 'Preliminaries'. It contains 1) the protocols for the evaluation of models and 2) datasets information, including images' size, numbers etc. We also provide similar information in Section 5.1 for pre-training. With the training details in each following sub-sections of results, readers should be able to reproduce the results easily.

---

### Decision · Action_Editors · 2023-02-27

**Recommendation:** Accept with minor revision

**Comment:**

The paper contributes a dataset that extends the existing Agriculture-Vision dataset by including 3,600 large, high-resolution, full-field, RGB, and near-infrared images. The paper then studies pre-training methods on the dataset, focusing on the momentum contrast framework, and integrates the Pixel-to-Propagation Module into it. This paper benchmarks these self-supervised pre-training methods on downstream classification and segmentation tasks, showing some useful insights and future research directions.

Four reviewers carefully review the paper. While they generally accept the paper contributions, they also raise critical concerns. A few common ones include 1) less clear writing, including some confusing math notations and typos, 2) the less clear motivation and description of temporal information and contrast in the dataset, 3) why to choose momentum contrastive learning methods for the study, rather than other self-supervised learning methods. The authors responded and revised the paper that makes the paper clearer. Reviewers are generally satisfied with these common issues.

In addition, Reviewer TPh3 raised issues about style inconsistency among a few figures, unclear description of the experimental setups, training/test protocols, etc, and less significant contributions and insights from the paper. The authors have carefully responded to these issues and revised the paper accordingly. The reviewer appreciates most of these responses.

After the discussion phase, two of the reviewers suggest acceptance while the others suggest not. AE follows up the discussions and would like to accept the paper given that 1) the releasing of this dataset is valuable for agricultural research, 2) while only momentum contrast based methods are studied, the studies are self consistent and generalizable to other self-training methods, 3) the paper and the dataset would advance the field by bringing features (imagery data, temporal contrast, fine-grained classification, etc) new to the community.

In the final version, the authors are strongly suggested to follow the suggestions from the reviewer and make writing of the data collection and training/testing protocol clear and self-sufficient, rather than to ask the readers to refer to the previous literature. Please also revise the format of citation and correct typos throughout the paper,

**Audience:**

Individuals working on the areas of semantic segmentation, remote sensing, and agricultural image analysis would be interested in findings contributed by this paper.

**Claims And Evidence:**

The released dataset, pre-training analysis by self-supervised learning methods, and usefulness for downstream tasks are genenrally supported by the expeirments and results.